   

# Melanocyte differentiation and mechanosensation are differentially modulated by distinct extracellular matrix proteins

Carole Luthold [1,2,3✉], Marie Didion[1,2,3], Vanessa Samira Rácz [1,2,3], Emilio Benedum [1,2,3], Ann-Kathrin Burkhart [1,2,3], Nina Demmerle[1,2,3], Evelyn Wirth[1,2,3], Gubesh Gunaratnam[4], Sudharshini Thangamurugan [5], Volkhard Helms[5], Markus Bischoff[4], Annika Ridzal[1,3] & Sandra Iden [1,2,3,6,7✉]

## Abstract

**Melanocyte dysfunctions can lead to pigmentation disorders or melanoma. Melanocytes interact context-dependently with various types of ECM, including collagens and fibronectin. Alterations in ECM composition and stiffness can impact cell behavior, but their specific roles for melanocyte functions remain unclear. We here exposed melanocytes to different ECM proteins and varying substrate stiffnesses, and identified MITF, a key regulator of melanocyte differentiation and function, as an ECM- and mechanosensitive transcription factor. Moreover, distinct ECM proteins and substrate stiffness engaged a FAK/MEK/ERK/MITF signaling axis to control melanocyte functions. Collagen I restricted FAK and ERK activation, promoting elevated nuclear MITF levels, melanocyte proliferation and a differentiated transcriptomic signature. Conversely, fibronectin elicited FAK and ERK activation, reduced nuclear MITF, increased motility and a dedifferentiated transcriptomic signature. On fibronectin, inhibiting MEK/ERK activity caused increased MITF nuclear localization and enhanced melanogenesis. Additionally, FAK inhibition reduced ERK activation and enhanced melanogenesis, supporting that FAK acts upstream of ERK. Finally, melanocytes show ECM-dependent mechanoresponses. In summary, extrinsic cues exert substantial effects on melanocyte function, involving ERK-dependent MITF regulation.**

**Keywords** Melanocyte Differentiation; MITF; Extracellular Matrix Cues; ERK Signaling; Mechanosensation
**Subject Categories** Cell Adhesion, Polarity & Cytoskeleton; Development

## Introduction

Melanocytes (MCs) are pigment-producing cells found in the basal layer of the skin epidermis, and in various organs such as the brain, heart, and eyes (Centeno et al, 2023). The pronounced dendritic morphology of cutaneous MCs enables an efficient melanosome transfer to surrounding keratinocytes, providing photoprotection (Benito-Martínez et al, 2021; Cui and Man, 2023; Domingues et al, 2020; Hirobe, 2014; Prospéri et al, 2024). Dysfunctions of MCs can cause pigmentation disorders such as albinism and vitiligo (Coutant et al, 2024), and can lead to malignant transformation into melanoma, an aggressive form of skin cancer responsible for most skin cancer-related deaths (Matthews et al, 2017).

The specification and differentiation of MCs from the neural lineage are governed by the melanocyte-inducing transcription factor (MITF). MITF regulates MC differentiation by controlling the expression of pigmentation genes (e.g., *Tyr*, *Trp1*, and *Trp2*) and maintains cellular homeostasis by modulating genes involved, among others, in cell cycle progression (e.g., *Cdk2*) and apoptosis (e.g., *Bcl2*) (Goding and Arnheiter, 2019; Kawakami and Fisher, 2017). MITF is also a major player in melanoma progression, influencing both the melanoma cell differentiation state and plasticity, contributing to high tumor heterogeneity. Within a single tumor, different cell states co-exist: highly differentiated, proliferative melanocytic cells, which are associated with high MITF levels, whereas slow-cycling, dedifferentiated, invasive stem-like cells correlate with low MITF levels (Arozarena and Wellbrock, 2019; Carreira et al, 2006; Cheli et al, 2011; Hoek et al, 2008; Müller et al, 2014; Popovic and Tartare-Deckert, 2022; Rambow et al, 2019; Tirosh et al, 2016; Tsoi et al, 2018; Wouters et al, 2020). Interestingly, differentiated MCs can give rise to melanoma through a process that includes their reprogramming and dedifferentiation (Köhler et al, 2017). However, the specific factors and mechanisms driving this dedifferentiation of mature MCs, and the role of MITF in these processes, remain poorly understood.

[1]Cell & Developmental Biology, Saarland University, Faculty of Medicine, Kirrberger Strasse, Homburg/Saar, Germany. [2]Center for Gender-specific Biology & Medicine (CGBM), Saarland University, Faculty of Medicine, Kirrberger Strasse, Homburg/Saar, Germany. [3]Center for Human & Molecular Biology (ZHMB), Saarland University, Saarbrücken, Germany. [4]Institute for Medical Microbiology and Hygiene, Kirrberger Strasse, Homburg/Saar, Germany. [5]Center for Bioinformatics, Saarland University, Saarbrücken, Germany. [6]PharmaScienceHub (PSH), Saarland University, Saarbrücken, Germany. [7]Center for Biophysics (ZBP), Saarland University, Saarbrücken, Germany. ✉E-mail: carole.luthold@uni-saarland.de; sandra.iden@uni-saarland.de

Therefore, understanding how MITF affects MC plasticity and identifying novel factors that regulate MITF expression and activity could provide new insights into the mechanisms underlying MC transformation.

In addition to the heterotypic cell-cell interaction with keratinocytes, MCs are in direct contact with the basement membrane, a specialized extracellular matrix (ECM) rich in type IV collagen (COL IV), which forms the junction with the underlying dermis, comprising type I collagen (COL I) and fibronectin (FN) (Pfisterer et al, 2021). This implies that under physiological conditions, COL IV is the predominant ECM type epidermal MCs are exposed to, whereas interactions with dermal COL I and FN can occur due to altered basement membrane integrity, for instance, following injury or chronic UV exposure (Amano, 2009, 2016; Fisher and Rittié, 2018; Iriyama et al, 2011, 2020; Yoshihisa et al, 2014). Within the skin, a wide range of stiffness has been reported for different compartments, ranging from lower kPa (dermis-like) to lower MPa (epidermis-like) values (Biggs et al, 2020; Feng et al, 2022; Graham et al, 2019). Importantly, the cellular microenvironment, including ECM components and tissue stiffness, significantly impacts cell fate and function (Bonnans et al, 2014; Guilak et al, 2009; Walma and Yamada, 2020). Cells adapt to these molecular and mechanical parameters by transmitting environmental signals to intracellular signal transduction pathways, including mitogen-activated protein kinase (MAPK)/extracellular signal-regulated kinase (ERK) pathways (Tan et al, 2023). Crucial for this process are focal adhesions (FAs), dynamic multi-protein complexes at the plasma membrane that link the ECM to the actin cytoskeleton via transmembrane receptors such as integrins. Notably, focal adhesion kinase (FAK), a key signaling protein within FAs, is activated through integrin engagement with the ECM, resulting in autophosphorylation and activation of downstream signaling pathways like MAPK/ERK (Paszek et al, 2005). In various cell systems, such signal activation downstream of ECM cues has been reported to affect the localization and activation levels of various transcription factors and coregulatory factors, such as Yes-associated protein (YAP), leading to gene expression changes that impact processes like proliferation, survival, or differentiation (Ishihara and Haga, 2022).

In melanoma cells, MAPK signaling components have recently been reported to negatively regulate MITF nuclear localization and activity in melanoma cells: rapidly accelerated fibrosarcoma (RAF) proteins, acting upstream of ERK, interact directly with MITF, causing its cytoplasmic retention and reduced transcriptional activity (Estrada et al, 2022), while MAPK/ERK signaling, in collaboration with glycogen synthase kinase 3 (GSK3), controls MITF nuclear export (Ngeow et al, 2018). The transcriptional co-activator YAP, known to translocate into the nucleus in response to ECM stiffening in various cell types (Cai et al, 2021; Dupont et al, 2011; Huang et al, 2022; Miskolczi et al, 2018), promotes MITF expression in uveal and cutaneous melanoma cells (Barbosa et al, 2023; Miskolczi et al, 2018) and serves as MITF cofactor in uveal melanoma (Barbosa et al, 2023). This indicates a dual role of YAP in the control of MITF in melanoma. However, whether and how ECM cues modulate MITF remains to be elucidated.

So far, studies have separately explored the roles of ECM type and substrate stiffness for MC biology. For example, Hara et al reported that COL IV stimulates dendricity in human MCs on glass substrates (considered as ultra-hard stiffness) (Hara et al, 1994),

while melanin production on stiff PDMS substrates (5.5 MPa) coated with laminin is higher compared to softer counterparts (from 50 kPa to 1.8 MPa) (Choi et al, 2014). However, the potential synergistic effects of substrate stiffness and ECM components on MC functions remain to be defined. Understanding such effects is particularly important since the ECM undergoes continuous remodeling that includes de novo synthesis and degradation (Pfisterer et al, 2021). Various factors, such as skin aging, inflammation, sunlight exposure, wound healing, and fibrosis, can promote ECM remodeling, potentially resulting in aberrant ECM modifications (Pfisterer et al, 2021), which may modify the presentation of ECM ligands and alter substrate stiffness. For instance, during wound healing, the ECM transitions from an FN-rich provisional matrix to a COL I–rich structure as the tissue repairs. Nevertheless, it is still insufficiently understood how substrate stiffness influences the cellular responses to specific ECM components, and, conversely, whether cells need to engage in particular ECM interactions in order to sense environmental stiffness features.

This study investigated whether and how substrate stiffness and ECM components act together to modulate MC functions. We compared the combined effects of various ECM proteins and substrate stiffness on MC behavior. Given that MITF plays a central role in integrating environmental signals to regulate key aspects of MC identity, function, and plasticity—and considering its known regulation by the MAPK/ERK pathway, which mediates ECM signal transduction—we hypothesized that MITF may serve as a key effector of ECM-dependent cues in MCs. By investigating how ECM composition and mechanical properties affect MITF activity, we aimed to uncover mechanisms through which the microenvironment shapes MC behavior and plasticity. Our results revealed that ECM components control MC differentiation and function via an FAK-MEK-ERK-MITF axis, with different ECM types determining the ability of MCs to perceive and respond to mechanical stimuli in their environment.

## Results

### ECM components differentially affect MC morphology, adhesion and migration

To assess how extracellular mechanical cues affect MC behavior, we employed PDMS substrates of varying stiffnesses: ≈45 kPa, defined here as soft (Fig. EV1A), ≈140 kPa as intermediate (Fig. EV1B) and ≈900 kPa as stiff (Fig. EV1C). This range was chosen to reflect the wide range of stiffness found within the skin and because increased environmental stiffness is associated with multiple pathologies (Diazzi et al, 2020; Pfisterer et al, 2021; Wang et al, 2023). To mimic both normal and altered ECM conditions, COL IV-, COL I-, and FN-coated substrates of varying stiffness were used for cultures of either primary (pMCs) or immortalized murine MCs (stably expressing membrane-targeted tandem dimer Tomato; iMCs, generated for this study), followed by molecular, cellular, and functional assays.

Interestingly, visual inspection in phase-contrast and fluorescence microscopy revealed distinct ECM-specific phenotypes, with COL I and FN eliciting opposing cellular responses. iMCs cultured on FN-coated substrates exhibited a more dendritic shape

compared to the mostly bi- or tripolar MCs grown on COL I (Fig. 1A). To examine this phenotype further, we performed a Sholl analysis (Figs. 1A–C and EV2A; Binley et al, 2014; Sholl, 1953). Quantification of the dendritic complexity of MCs (Fig. 1B) and the corresponding Sholl profile (Figs. 1C and EV2A) confirmed that most iMCs exposed to COL IV and FN displayed between four to seven dendrites, whereas MCs grown on COL I exhibited mostly two to three dendrites (Fig. 1B). Noteworthy, over the range of stiffnesses tested, MC dendricity was not impacted by substrate stiffness, irrespective of the ECM protein type used (Fig. EV2A).

Since cell spreading is regulated by coordinated changes in adhesions to ECM and cytoskeletal reorganization, we next examined the cell area as well as key cell adhesion parameters. Among these, FAK activity by means of its autophosphorylation (p-FAK) and quantification of the number of focal adhesions (FAs) per cell served as indicators of responses resulting from MC-ECM interactions. In both pMCs and iMCs, the smallest cell area was detected on COL I (Figs. 1D,E and EV3A) and correlated with the lowest p-FAK levels (Figs. 1D,F and EV3B), whereas the largest cell areas (Figs. 1D,E and EV3A) were found on FN, correlating with high p-FAK signals (Figs. 1D,F and EV3B). Substrate stiffness, however, seemed to have less influence on the cell area: apart from a slight increase from soft to intermediate stiffness in iMCs cultured on COL I, we did not note significant changes of the cell area across the different substrate stiffnesses, regardless of the ECM protein used (Figs. 1E and EV3A). In contrast, both p-FAK and the number of FAs per $\mu m^2$ exhibited a significant stiffness-mediated increase in MCs grown on FN-coated substrates (Figs. 1F,G and EV3B,C). Conversely, the overall low p-FAK levels in COL I-exposed iMCs even decreased with stiffness (Fig. 1F), highlighting that stiffness sensing in MCs depends on the ECM type.

Finally, since cell morphology and FAs are closely linked to cell movement, we investigated whether the distinct ECM molecules influence MC motility. Live-cell imaging of iMCs revealed that both the distance traveled (Fig. 2A–C) and velocity (Fig. 2D) of cell migration decreased on COL I compared to COL IV and FN, whereas FN-grown iMCs showed increased motility (Fig. 2A–C) and faster migration (Fig. 2D) relative to their COL IV-grown counterparts.

Together, these data show that COL I and FN trigger opposite morphological, adhesive and migratory phenotypes in MCs.

## COL I and FN have opposite effects on melanin production and MC proliferation

Given the observed role of ECM components for MC morphology and migration, we next investigated whether ECM cues influence other MC functions. As a hallmark task of MCs, we examined their melanin production by measuring both intracellular melanin contents and melanin released into the culture medium. iMCs grown on COL I-coated substrates exhibited the highest intra- and extracellular melanin content, while melanin levels on FN were lowest compared to COL I and COL IV (Fig. 3A,B). Notably, the expression of *Tyr*, an essential gene for melanogenesis, correlated with melanin production, exhibiting a higher expression in iMCs exposed to COL I than in those exposed to FN (Fig. 3C). Like iMCs, pMCs cultured on FN produced the least melanin on soft and intermediate stiffness (Fig. EV3D). For all ECM types tested, the highest values for melanin production were observed at

high stiffness (Figs. EV3D and 3A,B). Collectively, these data demonstrated that compared to COL I, FN restricts melanin production.

Using a BrdU assay, we next evaluated MC proliferation. While stiffness had no effect on MC proliferation in the presence of COL IV and FN, we observed a higher proportion of BrdU-positive iMCs when exposed to stiff COL I-coated substrates, indicating a synergistic effect of COL I and stiffness on MC proliferation (Fig. 3D). Among the three ECM types tested, FN exposure resulted in the lowest proliferation rates (Fig. 3D), a finding that could be further confirmed by Ki67 immunostaining (Fig. 3E), as well as live-cell imaging followed by quantification of the percentage of dividing cells (Fig. 3F). Overall, these experiments revealed a stiffness-dependent proliferative phenotype triggered by COL I, while FN appears to restrict cell division of MCs.

Together, our findings show that melanin production and *Tyr* expression are highest in MCs exposed to COL I and lowest in FN-grown MCs. Moreover, cell proliferation was lowest on FN, and stiffness-mediated increase in proliferation was only observed on COL I-coated substrates, suggesting that MC mechanoresponses depend on the ECM type.

## MITF nuclear localization is modulated by ECM molecules and substrate stiffness, but does not correlate with nuclear YAP

Our data indicate an ECM-dependent regulation of MC proliferation and melanin production, with COL I and FN driving opposite effects on these cellular functions. Considering the central role of the transcription factor MITF in the control of both melanogenesis and cell cycle progression (Goding and Arnheiter, 2019), and taking into account the observed increase of the MITF target gene *Tyr* on COL I (Fig. 3C), we next examined the impact of ECM cues on the protein levels and subcellular localization of MITF (Figs. 4A–C and EV3E). Although overall MITF levels were slightly higher on FN compared to COL I, nuclear MITF intensity showed a 1.5-fold increase on COL I compared to FN (Fig. 4A, B). When considering the percentage of nuclear MITF relative to total cellular levels, a 2.6-fold enrichment was observed in MCs exposed to COL I compared to those on FN (Figs. 4A,C and EV3E). These differences align with elevated melanin production on COL I and suggest that ECM components influence MITF subcellular distribution.

Furthermore, a stiffness-mediated increase of MITF nuclear localization was noted in MCs on COL I (Figs. 4B,C and EV3E), thus identifying MITF as an ECM- and mechanosensitive transcription factor. The contrasting phenotypes of MCs grown on COL I vs. FN were also evident when correlating nuclear MITF levels to the cell area (Fig. EV4A): while MCs grown on COL I were characterized by a low ratio of cell area to nuclear MITF compared to COL IV, FN-exposed MCs showed high ratios, i.e., a large cell area was associated with low MITF. Interestingly, this ratio decreased with substrate stiffness on COL I and FN but not on COL IV (Fig. EV4A), suggesting that changes in nuclear MITF are not linearly related to morphological changes in response to ECM. siRNA-mediated knockdown of MITF (Fig. EV4B) confirmed the principal role of MITF for melanin production on both COL I and FN (Fig. 4D).

Given that YAP is a transcriptional co-activator known to translocate into the nucleus in response to ECM stiffening in

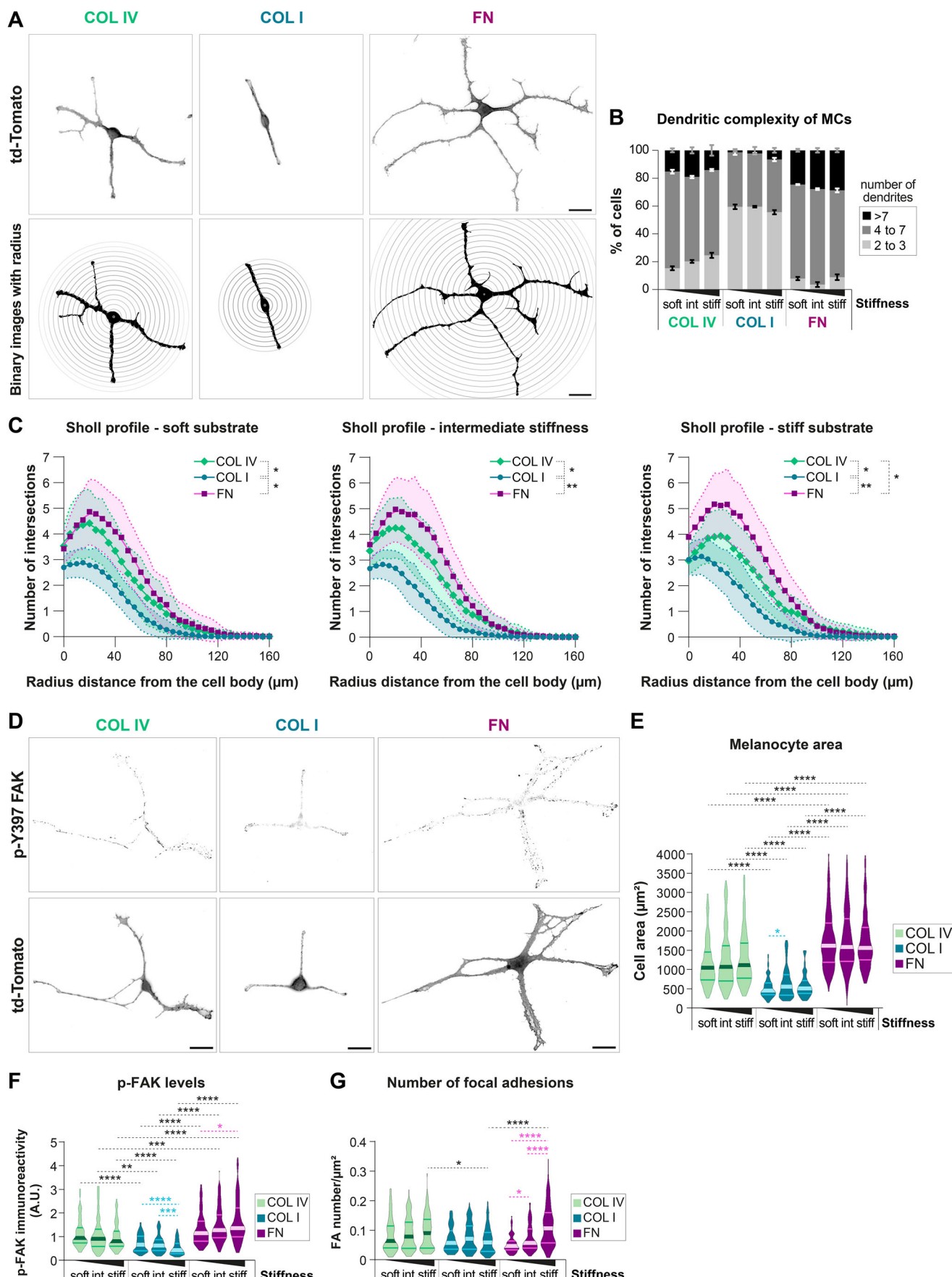

◀ **Figure 1. ECM components determine the morphology and adhesion of immortalized melanocytes.**

iMCs were cultured overnight on substrates of varying stiffness, defined here as soft (PDMS ratio 1:60 ≈ 40 kPa), intermediate (PDMS ratio 1:35 ≈ 130 kPa) and stiff (PDMS ratio 1:10 ≈ 1 MPa) coated with COL IV, COL I or FN. (A) Top: Representative micrographs of iMCs, expressing td-Tomato, cultured overnight on stiff substrates coated with COL IV, COL I or FN; scale bar: 20 μm. Bottom: Representative binary image of cells used for Sholl analysis, with radius drawn every 5 μm. (B) Quantification of the number of dendrites of iMCs; graph indicating the percentage of cells exhibiting 2 or 3 dendrites, between 4 and 7, or more than 7 dendrites; mean ± SEM, $N = 3$ (biological replicates), $n$(cells) ≥85. (C) Graphs showing the Sholl profile analysis, plotting the number of dendrite intersections against the distance from the cell body. The SEM are represented by the connecting curve (dotted line); $N = 3$ (biological replicates), $n$(cells) ≥85. Each curve represents an ECM type, and each graph represents a substrate stiffness condition; Welch ANOVA test: 1:60: *$p = 0.0101$ (COL I vs. FN), *$p = 0.024$ (COL I vs. COL IV), ns $p = 0.3119$ (FN vs. COL IV); 1:35: **$p = 0.0057$ (COL I vs. FN), *$p = 0.0142$ (COL I vs. COL IV), ns $p = 0.1536$ (FN vs. COL IV); 1:10: **$p = 0.0019$ (COL I vs. FN), *$p = 0.0138$ (COL I vs. COL IV), *$p = 0.0362$ (FN vs. COL IV). (D) Representative micrographs of iMCs (expressing membrane-targeted tandem dimer (td) Tomato) cultured on stiff substrates coated with COL IV, COL I or FN and immunostained for p-Y397-FAK and with DAPI; scale bar: 20 μm. (E) Quantification of the cell area of iMCs; violin plots display the medians and distributions of cell area in each condition; $N = 3$ (biological replicates), $n$(cells) ≥69; Kruskal–Wallis test: COL IV: ns $p > 0.9999$ (1:60 vs. 1:35), ns $p = 0.8765$ (1:60 vs. 1:10), ns $p > 0.9999$ (1:35 vs. 1:10); COL I: ****$p < 0.0001$ (1:60 vs. 1:35), ns $p = 0.2148$ (1:60 vs. 1:10), *$p = 0.234$ (1:35 vs. 1:10); FN: ns $p = 0.3094$ (1:60 vs. 1:35), ns $p > 0.9999$ (1:60 vs. 1:10), ns $p = 0.3022$ (1:35 vs. 1:10); 1:60: ****$p < 0.0001$ (COL I vs. FN), ****$p < 0.0001$ (COL I vs. COL IV), *$p = 0.0482$ (FN vs. COL IV); 1:35: ****$p < 0.0001$ (COL I vs. FN), **$p = 0.0033$ (COL I vs. COL IV), ns $p > 0.9999$ (FN vs. COL IV); 1:10: ****$p < 0.0001$ (COL I vs. FN), ****$p < 0.0001$ (COL I vs. COL IV), ns $p = 0.4259$ (FN vs. COL IV). (F) Quantification of p-FAK levels; violin plots show the medians and distributions of the integrated density of p-FAK per cell; $N ≥ 4$ (biological replicates), $n$(cells) ≥82; Kruskal–Wallis test: COL IV: ns $p = 0.4894$ (1:60 vs. 1:35), ns $p = 0.0928$ (1:60 vs. 1:10), ns $p > 0.9999$ (1:35 vs. 1:10); COL I: ns $p = 0.3868$ (1:60 vs. 1:35), ****$p < 0.0001$ (1:60 vs. 1:10), ****$p < 0.0001$ (1:35 vs. 1:10); FN: ns $p = 0.282$ (1:60 vs. 1:35), *$p = 0,0359$ (1:60 vs. 1:10), ns $p = 0.8607$ (1:35 vs. 1:10); 1:60: *$p = 0.0256$ (COL I vs. FN), ns $p = 0.0604$ (COL I vs. COL IV), ns $p > 0,9999$ (FN vs. COL IV); 1:35: ****$p < 0.0001$ (COL I vs. FN), *$p = 0.0427$ (COL I vs. COL IV), *$p = 0.0311$ (FN vs. COL IV); 1:10: ****$p < 0.0001$ (COL I vs. FN), ****$p < 0.0001$ (COL I vs. COL IV), ***$p = 0.0005$ (FN vs. COL IV). (G) Quantification of the number of focal adhesions per μm² per cell with violin plots displaying medians and distributions; $N ≥ 3$ (biological replicates), $n$(cells) ≥65; Kruskal–Wallis test: COL IV: ns $p > 0.9999$ (1:60 vs. 1:35), ns $p = 0.9173$ (1:60 vs. 1:10), ns $p > 0.9999$ (1:35 vs. 1:10); COL I: ns $p > 0.9999$ (1:60 vs. 1:35), ns $p > 0.9999$ (1:60 vs. 1:10), ns $p = 0.375$ (1:35 vs. 1:10); FN: *$p = 0.0363$ (1:60 vs. 1:35), ****$p < 0.0001$ (1:60 vs. 1:10), ****$p < 0.0001$ (1:35 vs. 1:10); 1:60: ns $p > 0,9999$ (COL I vs. FN), ns $p > 0.9999$ (COL I vs. COL IV), ns $p = 0.0733$ (FN vs. COL IV); 1:35: ns $p > 0.9999$ (COL I vs. FN), ns $p > 0.9999$ (COL I vs. COL IV), ns $p > 0.9999$ (FN vs. COL IV); 1:10: ****$p < 0.0001$ (COL I vs. FN), *$p = 0.017$ (COL I vs. COL IV), ns $p > 0.9999$ (FN vs. COL IV). AU arbitrary units. Source data are available online for this figure.

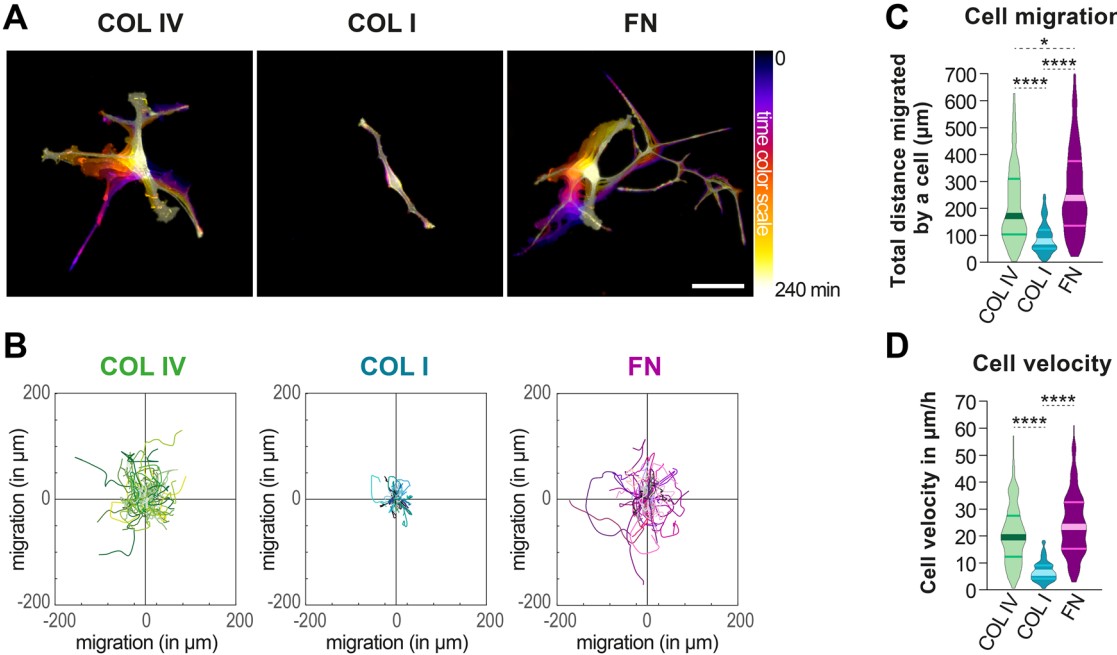

**Figure 2. Differential effects of ECM proteins on iMC motility, with COL I decreasing and FN increasing motility.**

(A) Image showing the temporal color code analysis from live-cell imaging of iMCs cultured overnight in a plastic chamber coated with COL IV, COL I, or FN to visualize cell dynamics over time. The different colors represent morphology and position of the iMCs at a given time (20 min intervals), with the color gradient indicating time progression (from 0 to 240 min); scale bar: 50 μm. (B) Tracks showing the path and distance of cells (one cell = one track) relative to the point of origin (time point zero) in the x and y plane (representative of one experiment). (C) Quantification of total distance traveled by iMCs; violin plots show the medians and distributions; $N = 3$ (biological replicates), $n$(cells) ≥113; Kruskal–Wallis test: ****$p < 0.0001$ (COL I vs. FN), ****$p < 0.0001$ (COL I vs. COL IV), *$p = 0.0239$ (FN vs. COL I). (D) Quantification of mean speed of iMCs; violin plots show the medians and distributions $N = 3$ (biological replicates), $n$(cells) ≥113; Kruskal–Wallis test: ****$p < 0.0001$ (COL I vs. FN), ****$p < 0.0001$ (COL I vs. COL IV), ns $p = 0.0792$ (FN vs. COL IV). Source data are available online for this figure.

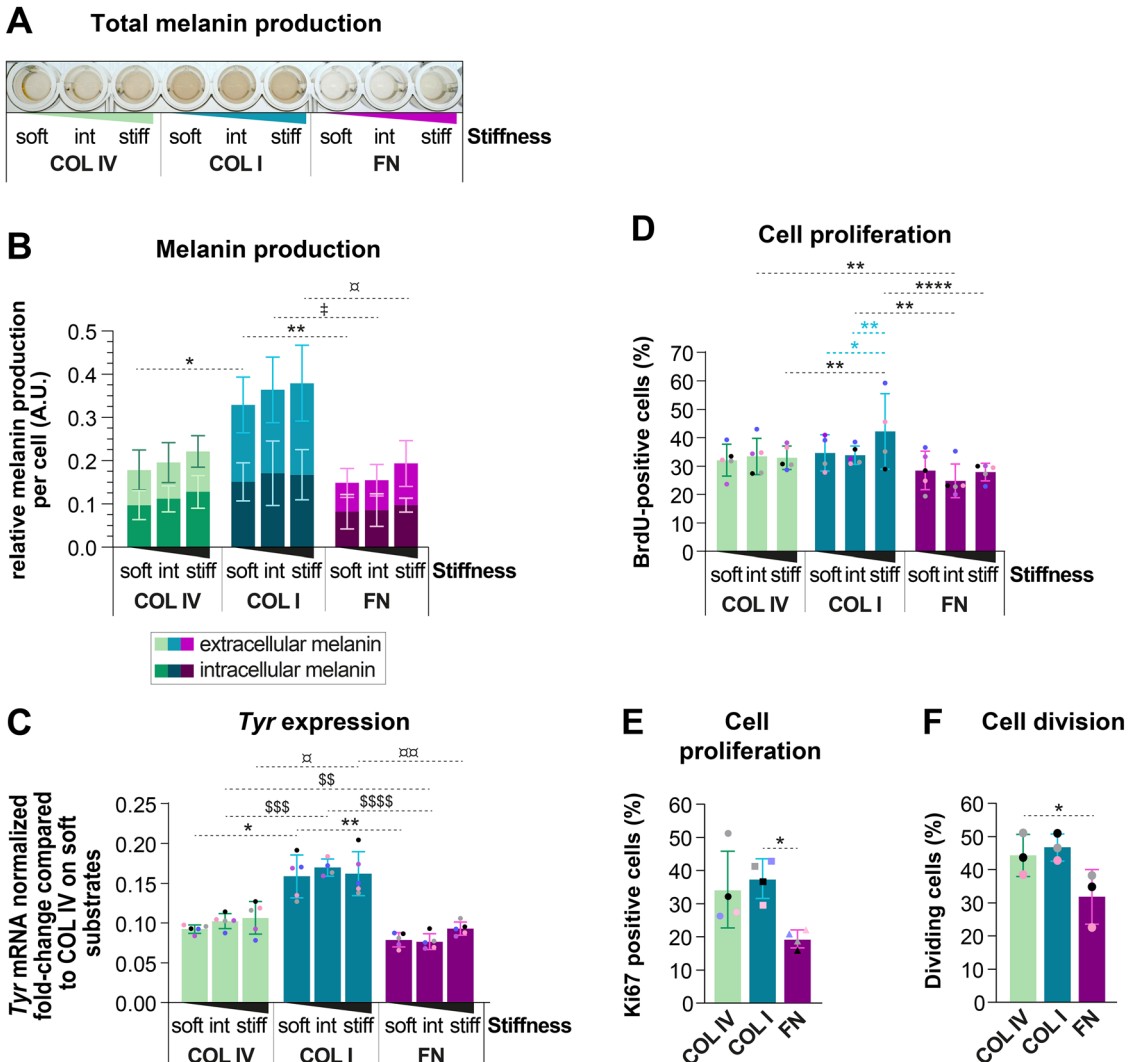

**Figure 3. COL I stimulates melanin production, and FN counteracts iMC proliferation.**

(A) Representative example of total melanin produced by iMCs cultured for 72 h on substrates of varying stiffness (soft, intermediate or stiff) coated with COL IV, COL I, or FN. (B) Quantification of intra- and extracellular melanin content by spectrophotometry at 405 nm from iMCs cultured as in (A); means ± SD, N ≥ 5 (biological replicates); Welch ANOVA test: COL IV: ns p = 0.9634 (1:60 vs. 1:35), ns p = 0.6945 (1:60 vs. 1:10), ns p = 0.9057 (1:35 vs. 1:10); COL I: ns p = 0.9214 (1:60 vs. 1:35), ns p = 0.7689 (1:60 vs. 1:10), ns p = 0.9936 (1:35 vs. 1:10); FN: ns p = 0.994 (1:60 vs. 1:35), ns p = 0.3219 (1:60 vs. 1:10), ns p = 0.4965 (1:35 vs. 1:10); 1:60: **p = 0.0033 (COL I vs. FN), *p = 0.0206 (COL I vs. COL IV), ns p = 0.7957 (FN vs. COL IV); 1:35: *p = 0.0154 (COL I vs. FN), ns p = 0.0517 (COL I vs. COL IV), ns p = 0.6287 (FN vs. COL IV); 1:10: *p = 0.0193 (COL I vs. FN), ns p = 0.0515 (COL I vs. COL IV), ns p = 0.8636 (FN vs. COL IV). (C) Quantification of *Tyr* mRNA expression of iMCs cultured overnight on substrates of varying stiffness (soft, intermediate or stiff) coated with COL IV, COL I, or FN; means ± SD, N = 5 (biological replicates); Kruskal–Wallis test: COL IV: ns p = 0.221 (1:60 vs. 1:35), ns p = 0.4313 (1:60 vs. 1:10), ns p = 0.9644 (1:35 vs. 1:10); COL I: ns p = 0.7861 (1:60 vs. 1:35), ns p = 0.9965 (1:60 vs. 1:10), ns p = 0.9095 (1:35 vs. 1:10); FN: ns p = 0.9733 (1:60 vs. 1:35), ns p = 0.0926 (1:60 vs. 1:10), ns p = 0.0636 (1:35 vs. 1:10); 1:60: *p = 0.0144 (COL IV vs. COL I), ns p = 0.0722 (COL IV vs. FN), **p = 0.004 (COL I vs. FN); 1:35: ***p = 0.0002 (COL IV vs. COL I), **p = 0.0089 (COL IV vs. FN), ****p < 0.0001 (COL I vs. FN); 1:10: *p = 0.0241 (COL IV vs. COL I), ns p = 0.4891 (COL IV vs. FN), **p = 0.0083 (COL I vs. FN). (D) Percentage of BrdU-positive cells from iMCs cultured 48 h on substrates of varying stiffness (soft, intermediate or stiff) coated with COL IV, COL I, or FN; means ± SD, N ≥ 4 (biological replicates); ordinary two-way ANOVA: COL IV: ns p = 0.9984 (1:60 vs. 1:35), ns p > 0.9999 (1:60 vs. 1:10), ns p > 0.9999 (1:35 vs. 1:10); COL I: ns p > 0.9999 (1:60 vs. 1:35), *p = 0.0296 (1:60 vs. 1:10), **p = 0.0063 (1:35 vs. 1:10); FN: ns p = 0.6252 (1:60 vs. 1:35), ns p > 0.9999 (1:60 vs. 1:10), ns p = 0.7993 (1:35 vs. 1:10); 1:60: ns p = 0.0965 (COL I vs. FN), ns p = 0.9351 (COL I vs. COL IV), ns p = 0.6423 (FN vs. COL IV); 1:35: **p = 0.0014 (COL I vs. FN), ns p > 0.9999 (COL I vs. COL IV), **p = 0.0025 (FN vs. COL IV); 1:10: ****p < 0.0001 (COL I vs. FN), **p = 0.0019 (COL I vs. COL IV), ns p = 0.2238 (FN vs. COL IV). (E) Percentage of Ki67-positive iMCs cultured 48 h on stiff substrates; means ± SD, N = 4 (biological replicates); Welch ANOVA test: ns p = 0.9366 (COL I vs. FN), ns p = 0.1935 (COL I vs. COL IV), *p = 0.0128 (FN vs. COL IV). (F) Percentage of dividing cells from live-cell imaging of iMCs cultured overnight on COL IV, COL I, and FN; means ± SD, N = 3 (biological replicates); RM one-way ANOVA test: ns p = 0.0913 (COL I vs. FN), ns p = 0.7956 (COL I vs. COL IV), *p = 0.0471 (FN vs. COL IV). AU arbitrary units. Source data are available online for this figure.

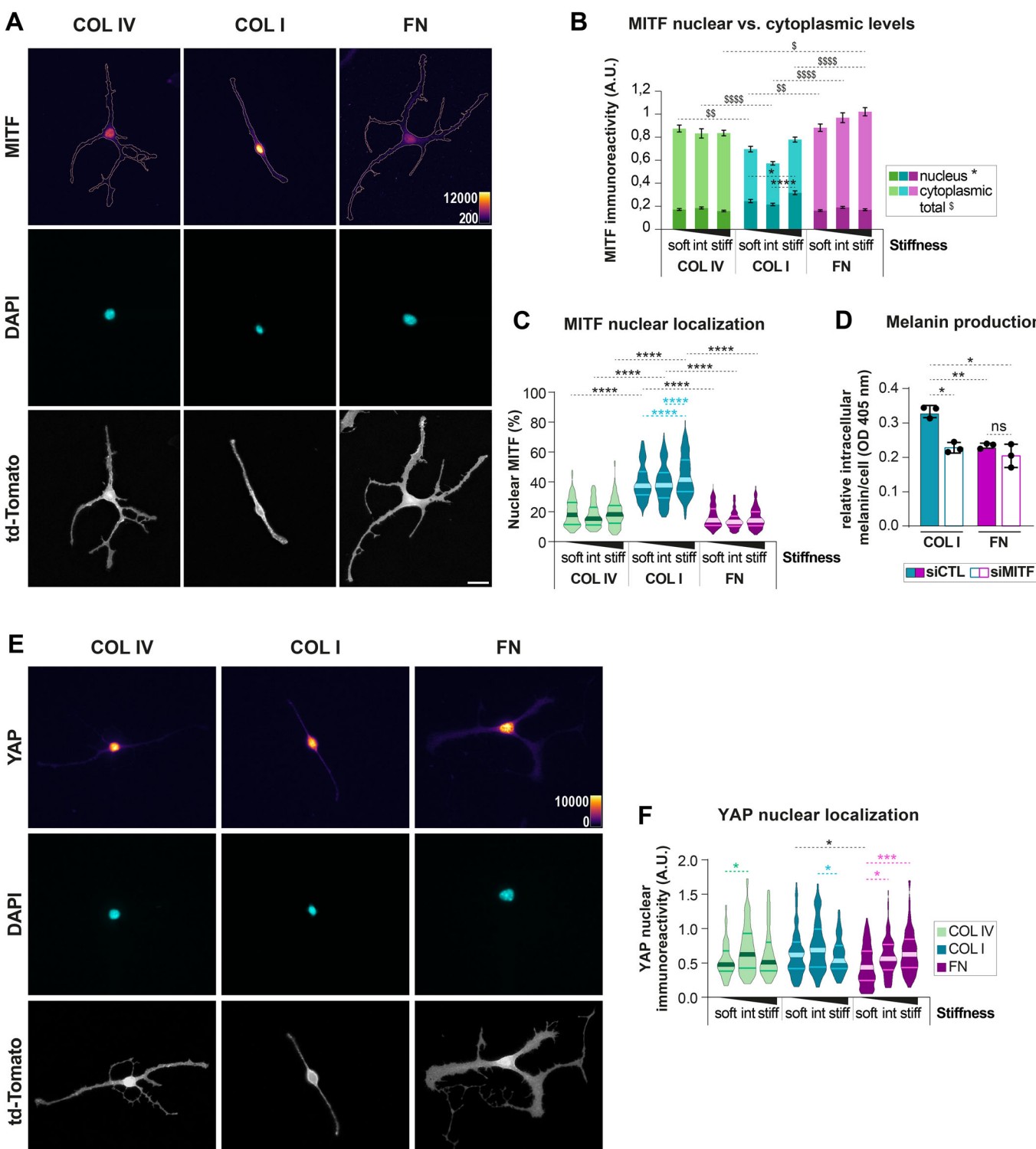

various cell types, including melanoma cells (Miskolczi et al, 2018), and that YAP may serve as a potential MITF partner in uveal melanoma cells (Barbosa et al, 2023), we wondered whether YAP could be associated with the observed ECM-dependent MC phenotypes. We examined YAP subcellular localization and, consistent with previous reports, observed a stiffness-mediated increase in nuclear YAP, with the strongest response at intermediate stiffness levels on COL IV and COL I, and on stiff FN-coated substrates (Fig. 4E,F). Importantly, however, unlike nuclear MITF, we hardly observed significant differences in nuclear YAP when comparing the three ECM types, making it unlikely that MCs utilize YAP for ECM-specific melanin production.

**Figure 4. MITF nuclear localization is modulated by both ECM molecules and substrate stiffness.**

(A) Representative micrographs of iMCs expressing td-Tomato cultured overnight on stiff substrates coated with COL IV, COL I or FN and stained for MITF and DAPI; scale bar: 20 μm. A pseudo-color intensity scale ('fire' color map) was applied to enhance visualization of MITF nuclear localization. (B) Quantification of cytoplasmic and nuclear MITF levels; graphs show the mean ± SEM of nuclear and cytoplasmic MITF intensities per cell 24 h post-plating on soft, intermediate, or stiff substrates coated with COL IV, COL I, or FN; $N \geq 3$ (biological replicates), $n$(cells) ≥89; Kruskal–Wallis test for nuclear MITF: COL IV: ns $p = 0.5977$ (1:60 vs. 1:35), ns $p > 0.9999$ (1:60 vs. 1:10), ns $p = 0.1299$ (1:35 vs. 1:10); COL I: ns $p = 0.5303$ (1:60 vs. 1:35), *$p = 0.0156$ (1:60 vs. 1:10), ****$p < 0.0001$ (1:35 vs. 1:10); FN: ns $p = 0.1425$ (1:60 vs. 1:35), ns $p > 0.9999$ (1:60 vs. 1:10), ns $p = 0.1758$ (1:35 vs. 1:10); 1:60: ***$p = 0.0004$ (COL I vs. FN), **$p = 0.0012$ (COL I vs. COL IV), ns $p > 0.9999$ (FN vs. COL IV); 1:35: ns $p > 0.9999$ (COL I vs. FN), ns $p > 0.9999$ (COL I vs. COL IV), ns $p > 0.9999$ (FN vs. COL IV); 1:10: ****$p < 0.0001$ (COL I vs. FN), ****$p < 0.0001$ (COL I vs. COL IV), ns $p > 0.9999$ (FN vs. COL IV); Kruskal–Wallis test for total MITF: COL IV: ns $p = 0.5921$ (1:60 vs. 1:35), ns $p > 0.9999$ (1:60 vs. 1:10), ns $p = 0.9814$ (1:35 vs. 1:10); COL I: \$ $p = 0.0122$ (1:60 vs. 1:35), ns $p = 0.1163$ (1:60 vs. 1:10), $p < 0.0001$ (1:35 vs. 1:10); FN: ns $p = 0.9309$ (1:60 vs. 1:35), \$\$ $p = 0.0067$ (1:60 vs. 1:10), ns $p = 0.1094$ (1:35 vs. 1:10); 1:60: \$\$ $p = 0.002$ (COL I vs. FN), \$\$ $p = 0.0055$ (COL I vs. COL IV), ns $p > 0.9999$ (FN vs. COL IV); 1:35: \$\$\$\$ $p < 0.0001$ (COL I vs. FN), \$\$\$\$ $p < 0.0001$ (COL I vs. COL IV), ns $p > 0.9999$ (FN vs. COL IV); 1:10: \$\$\$\$ $p < 0.0001$ (COL I vs. FN), ns $p > 0.9999$ (COL I vs. COL IV), \$ $p = 0.0272$ (FN vs. COL IV). (C) Quantification of nuclear MITF levels; violin plots show the medians and distributions of the percentage of nuclear MITF per cell; $N \geq 3$ (biological replicates), $n$(cells) ≥67; Kruskal–Wallis test: COL IV: ns $p = 0.5743$ (1:60 vs. 1:35), ns $p > 0.9999$ (1:60 vs. 1:10), ns $p = 0.8231$ (1:35 vs. 1:10); COL I: ns $p > 0.9999$ (1:60 vs. 1:35), *$p = 0.0305$ (1:60 vs. 1:10), **$p = 0.004$ (1:35 vs. 1:10); FN: ns $p = 0.0601$ (1:60 vs. 1:35), ns $p > 0.9999$ (1:60 vs. 1:10), ns $p = 0.5688$ (1:35 vs. 1:10); 1:60: ****$p < 0.0001$ (COL I vs. FN), ****$p < 0.0001$ (COL I vs. COL IV), ns $p > 0.9999$ (FN vs. COL IV); 1:35: ****$p < 0.0001$ (COL I vs. FN), ****$p < 0.0001$ (COL I vs. COL IV), ns $p = 0.8645$ (FN vs. COL IV); 1:10: ****$p < 0.0001$ (COL I vs. FN), ****$p < 0.0001$ (COL I vs. COL IV), ns $p = 0.4252$ (FN vs. COL IV). (D) Quantification of intracellular melanin levels following siRNA-mediated MITF knockdown. iMCs were transfected with control siRNA (siCtrl) or MITF-targeting siRNA (siMITF) 4 h after plating on COL I or FN. Seventy-two hours post-transfection, cells were harvested and melanin content was measured by spectrophotometry at 405 nm; means ± SD, $N = 3$ (biological replicates); Welch ANOVA test: **$p = 0.0062$ (COL I - siCTL vs. FN-siCTL), *$p = 0.0108$ (COL I - siCTL vs. COL I – siMITF), *$p = 0.0368$ (COL I - siCTL vs. FN – siMITF), ns $p = 0.9827$ (FN - siCTL vs. COL I – siMITF), ns $p = 0.8274$ (FN - siCTL vs. FN – siMITF), ns $p = 0.671$ (COL I - siMITF vs. FN – siMITF). (E) Representative micrographs of iMCs expressing td-Tomato cultured overnight on stiff substrates coated with COL IV, COL I, or FN and stained for YAP and DAPI; scale bar: 20 μm. A pseudo-color intensity scale ("fire" color map) was applied to enhance visualization of YAP nuclear localization. (F) Quantification of nuclear YAP levels; violin plots show the medians and distributions of the nuclear integrated density of YAP per cell; $N = 3$ (biological replicates), $n$(cells) ≥71; Kruskal–Wallis test: COL IV: *$p = 0.0147$ (1:60 vs. 1:35), ns $p = 0.3972$ (1:60 vs. 1:10), ns $p = 0.4846$ (1:35 vs. 1:10); COL I: ns $p = 0.1505$ (1:60 vs. 1:35), ns $p > 0.9999$ (1:60 vs. 1:10), *$p = 0.0485$ (1:35 vs. 1:10); FN: *$p = 0.0112$ (1:60 vs. 1:35), ***$p = 0.0004$ (1:60 vs. 1:10), ns $p = 0.8248$ (1:35 vs. 1:10); 1:60: *$p = 0.0198$ (COL I vs. FN); ns $p > 0.9999$ (COL I vs. COL IV), ns $p > 0.9999$ (FN vs. COL IV); 1:35: ns $p = 0.6123$ (COL I vs. FN), ns $p > 0.9999$ (COL I vs. COL IV), ns $p > 0.9999$ (FN vs. COL IV); 1:10: ns $p > 0.9999$ (COL I vs. FN), ns $p > 0.9999$ (COL I vs. COL IV), ns $p > 0.9999$ (FN vs. COL IV). AU arbitrary units, td-Tomato membrane-targeted tandem dimer Tomato. Source data are available online for this figure.

Collectively, our findings suggest that ECM subtypes regulate MC behavior and function likely through modulation of MITF localization and activity, independently of YAP.

## Bulk RNA sequencing confirms that ECM types elicit significant changes in the expression of genes associated with MC pigmentation and differentiation

Our data so far indicate that ECM composition modulates MITF localization and activity in MCs, suggesting that ECM components may influence MITF-regulated transcriptional programs. To investigate this further and gain a broader understanding of how ECM components shape the MC phenotype, we performed bulk RNA sequencing (RNAseq) on iMCs cultured on stiff substrates coated with COL IV, COL I, or FN (Figs. 5 and EV5). To capture global transcriptomic changes associated with distinct ECM environments and to identify gene signatures linked to MC differentiation, plasticity, and other relevant pathways, we analyzed the expression of MITF target genes, MC-specific genes, and broader signaling networks that are associated with the phenotypic shifts observed across ECM conditions.

Comparison of global transcriptomic profiles across the ECM types used revealed a range of differentially expressed genes (DEGs) (Figs. 5 and EV5B,C). Consistent with the reduced melanin production observed for MCs on FN (Fig. 3B), several MITF-dependent target genes, which are also pigmentation-related (e.g., *Dct*, *Pmel*, *Tyr*, and *Gpr143*), were significantly downregulated on FN compared to COL I (Figs. 5A,B and EV5D). In addition, further components involved in melanin biosynthesis and trafficking—such as *Oca2*, *Adcy2*, and *Rab27a*—also showed reduced expression on FN (Fig. 5A,B). Of note, some of the MITF-dependent DEGs have previously been linked to apoptosis (*Trpm1*) and cell cycle

progression (*Ccnd1*, *Cdkn1a*) (Fig. 5A). Global gene expression analysis indeed revealed further DEGs related to cell cycle control and apoptosis (Fig. 5C,D), in line with the observed decrease in proliferation of MCs cultured on FN when compared to COL I (Fig. 3D,E). In addition, FN elicited differential expression of genes associated with cellular senescence (Fig. 5E). Taken together, these findings suggest that adhesion to FN promotes a transcriptional program in MCs that may counteract cell cycle progression.

Considering that MITF-dependent gene expression overlapped with transcriptional programs involved in melanocyte development, and that these programs were downregulated in MCs cultured on FN (Fig. 5A), we extended our analysis to gene expression signatures associated with melanoma plasticity and differentiation states (Arozarena and Wellbrock, 2019; Durand et al, 2024; Huang et al, 2021; Konieczkowski et al, 2014; Pessoa et al, 2021; Rambow et al, 2015, 2019; Tsoi et al, 2018). Comparing FN to COL I revealed downregulation of genes associated with melanocytic differentiation on FN, such as *Apoe*, next to the previously mentioned MITF target genes (e.g., *Dct*, *Pmel*, *Tyr*, *Trpm1*, *Gpr143*, and *Irf4*) and melanin biosynthesis-related genes (e.g., *Oca2*, *Adcy2*) (Fig. 5A,B,F). However, core melanocytic factors such as *Sox10*, *Lef1*, *Creb1*, *Pax3*, and *Mlana* (Fig. 5F) were not significantly altered between COL I and FN, suggesting maintenance of lineage identity. In contrast, several genes associated with cell plasticity (*Tead1*, *Fzd4*, and *Tagln2*) and neural crest-like features (*Ngfr* and *Erbb3*) were significantly upregulated on FN relative to COL I (Fig. 5G). Additionally, genes previously associated with a dedifferentiated state, such as *Axl*, *Itga8*, *Timp1*, *Hmga1*, and *Nkd2*, were upregulated on FN compared to COL I (Fig. 5H). Finally, gene set enrichment analysis (GSEA) identified "nervous system development" as the top enriched Reactome pathway in iMCs cultured on FN compared to COL I (Fig. EV5E),

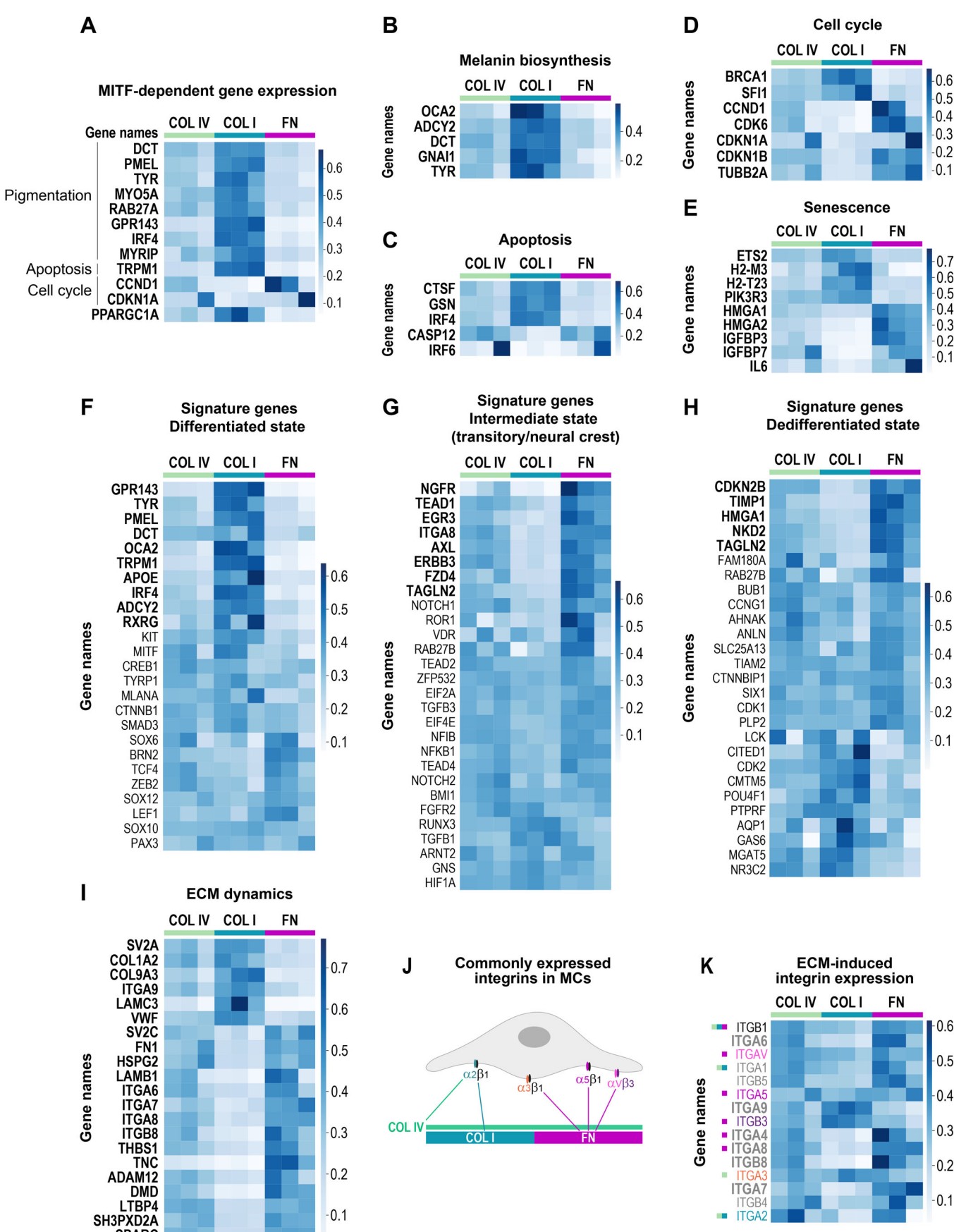

**Figure 5. ECM-dependent transcriptional changes in iMCs.**

iMCs were cultured overnight on stiff substrates coated with COL IV, COL I, or FN. Gene expression heatmaps showing transcript abundance across ECM conditions; $N = 3$ (biological replicates). Darker blue indicates higher expression; lighter blue indicates lower expression; bold labels indicate statistically differentially expressed genes ($p$ values derived from Wald tests in DESeq2 with Benjamini–Hochberg adjustment). Displayed values represent normalized expression levels per gene across conditions (e.g., standard scaling), enabling comparison of the same gene between ECM conditions despite differences in expression range. Panels show: (A) differentially expressed MITF-dependent genes; (B) genes involved in melanin biosynthesis; (C–E) genes related to apoptosis, cell cycle, and senescence, respectively; (F–H) signature genes of the differentiated, intermediate, and dedifferentiated states; genes were grouped by clustering of expression profiles; (I) genes involved in ECM dynamics; (J) schematic overview of integrins commonly expressed in MCs; and (K) integrin gene expression across COL IV, COL I, and FN conditions. Source data are available online for this figure.

further supporting the emergence of a phenotype characterized by increased plasticity and reduced differentiation on FN.

Gene set enrichment analysis (GSEA) identified "ECM organization" and "Degradation of ECM" as the top enriched Reactome pathway in iMCs cultured on FN compared to COL I (Fig. EV5E), indicating transcriptional remodeling of ECM-associated genes in response to ECM molecules. Among these, collagen genes (*Col1a2* and *Col9a3;* Fig. 5I) were downregulated on FN, while non-collagenous ECM components, including *Fn1, Tnc, Hspg2,* and *Thbs1*, showed increased expression (Fig. 5I). In addition, ECM-modifying genes (*Sparc, Adam12, Serpine1;* Fig. 5I) were also upregulated, supporting a potential active ECM remodeling on FN. Together, these gene expression patterns align with the emergence of a cell state characterized by increased migratory potential and plasticity in MCs grown on FN.

Interestingly, several integrin subunits were also differentially expressed across ECM conditions (Fig. 5I). A broad repertoire of integrins was found to be expressed in iMCs grown on COL IV, COL I, and FN, including subunits commonly reported in MCs (Arias-Mejias et al, 2020; Hara et al, 1994; Morelli et al, 1993; Pinon and Wehrle-Haller, 2011; Scott et al, 1992), such as *Itga2, Itga3, Itga4, Itga5, ItgaV, Itgb1,* and *Itgb3*, which mediate adhesion to collagens and fibronectin (Fig. 5J,K). While most of these were expressed independently of the ECM type used, we also noted that *Itga6, Itga7, Itga8,* and *Itgb8* were upregulated on FN, while *Itga9* was upregulated on COL I (Fig. 5I). Taken together, our data suggests that while ECM composition influences integrin gene expression to some extent, changes in integrin profiles alone may not fully account for the phenotypic differences observed between the distinct ECM components.

## ECM-dependent FAK/MEK/ERK activation controls MITF localization and activity, as well as melanogenesis

Given the ECM-dependent phenotypic shifts in MCs, we next analyzed expression changes in key signaling pathways— PI3K/Akt (Larribere et al, 2004; Phung et al, 2011; Shi et al, 2016; Wang et al, 2016), Wnt (Colombo et al, 2022; Katkat et al, 2023; Sinnberg et al, 2018), and MAPK (Buscà and Ballotti, 2000; Estrada et al, 2022; Ngeow et al, 2018; Ostojić et al, 2021; Wellbrock and Arozarena, 2015)— known to regulate MITF activity, melanocyte differentiation, and plasticity. While only a few PI3K/Akt-associated genes were differentially expressed, with *Brca1* and *Sgk2* downregulated and *Sgk1* upregulated on FN (Fig. 6A), Wnt signaling showed a broader modulation (Fig. 6B). Among the DEGs, multiple components—including the canonical target *Ccnd1*, the positive regulators *Frat2, Fzd4,* and *Ccdc88c*, and the feedback inhibitor

*Nkd2*—were upregulated on FN, suggesting an activation of both canonical and non-canonical Wnt signaling pathways, potentially accompanied by feedback regulation (Fig. 6B). The most pronounced transcriptional changes, however, were observed in MAPK (Fig. 6C) and ERK signaling (Fig. 6D), reflected by a high number of DEGs associated with these pathways. Some upstream activators of ERK (*Cnksr1, Rasgrp3, Pak3*) were downregulated on FN compared to COL I (Fig. 6C). In contrast, genes enhancing ERK signaling (*Igf2bp1, Irak2, Il1rap,* and *Ngfr*) and transcriptional ERK targets (*Etv4, Il6, Tnc, Lif,* and *Col1a2*) were upregulated on FN-coated substrates (Fig. 6C,D). In addition, upstream modulators of ERK, such as *Erbb3, Axl,* and *Itga6*, were also upregulated (Fig. 6C,D).

These data pointed to ECM-driven remodeling of MAPK/ERK signaling, prompting us to directly assess ERK activation in iMCs grown on COL IV, COL I, and FN. Immunoblot analyses of ERK1/2 phosphorylation (p-ERK) revealed an ECM-dependent modulation of ERK activity, with FN-grown iMCs featuring highest p-ERK levels, and COL I-exposed MCs displaying lowest p-ERK levels (Fig. 6E). Considering these differential p-ERK signals observed with the various ECM types and given that in melanoma cells, MAPK/ERK has recently been shown to negatively control MITF nuclear localization and activity (Estrada et al, 2022; Ngeow et al, 2018), we next asked whether MEK/ERK activation was causal for the observed ECM-dependent MC responses. For this, we used Trametinib, a pharmacological inhibitor targeting MEK, the upstream kinase mediating ERK phosphorylation and activation (Gilmartin et al, 2011). Next to the expected loss of ERK phosphorylation (Fig. EV6A), Trametinib treatment (indicated as MEKi) increased *Tyr* mRNA expression (Fig. 6F) and resulted in increased nuclear MITF in iMCs exposed to COL IV and FN, reaching the nuclear MITF levels observed on COL I (Fig. 6G,H). Congruent with this, melanin production by iMCs cultured on both COL IV- and FN-coated substrates and treated with Trametinib were significantly elevated towards those levels observed in COL I conditions (Fig. 6I). Similar results were obtained using an ERK inhibitor Ravoxertinib (Blake et al, 2016; Varga et al, 2020) (Fig. 6J–L). This indicates that MEK/ERK activity in MCs negatively regulates nuclear MITF levels. We next aimed to delineate signals that might transduce ECM sensing to activation of ERK. Given the correlation between elevated FAK and ERK activity in FN-exposed MCs and considering that FAK has been reported to transduce ECM signals through MAPK/ERK (Paszek et al, 2005), we inhibited FAK in iMCs. As expected, treatment of iMCs with the FAK inhibitor Ifebemtinib (Li et al, 2021) resulted in reduced p-FAK levels as well as a reduced number of FAs in MCs (Fig. EV6B, C). More importantly, FAK inhibition resulted in a reduction of p-ERK levels as assessed by immunocytochemistry (Fig. 6M,N) and caused a significant increase in melanin production (Fig. 6O).

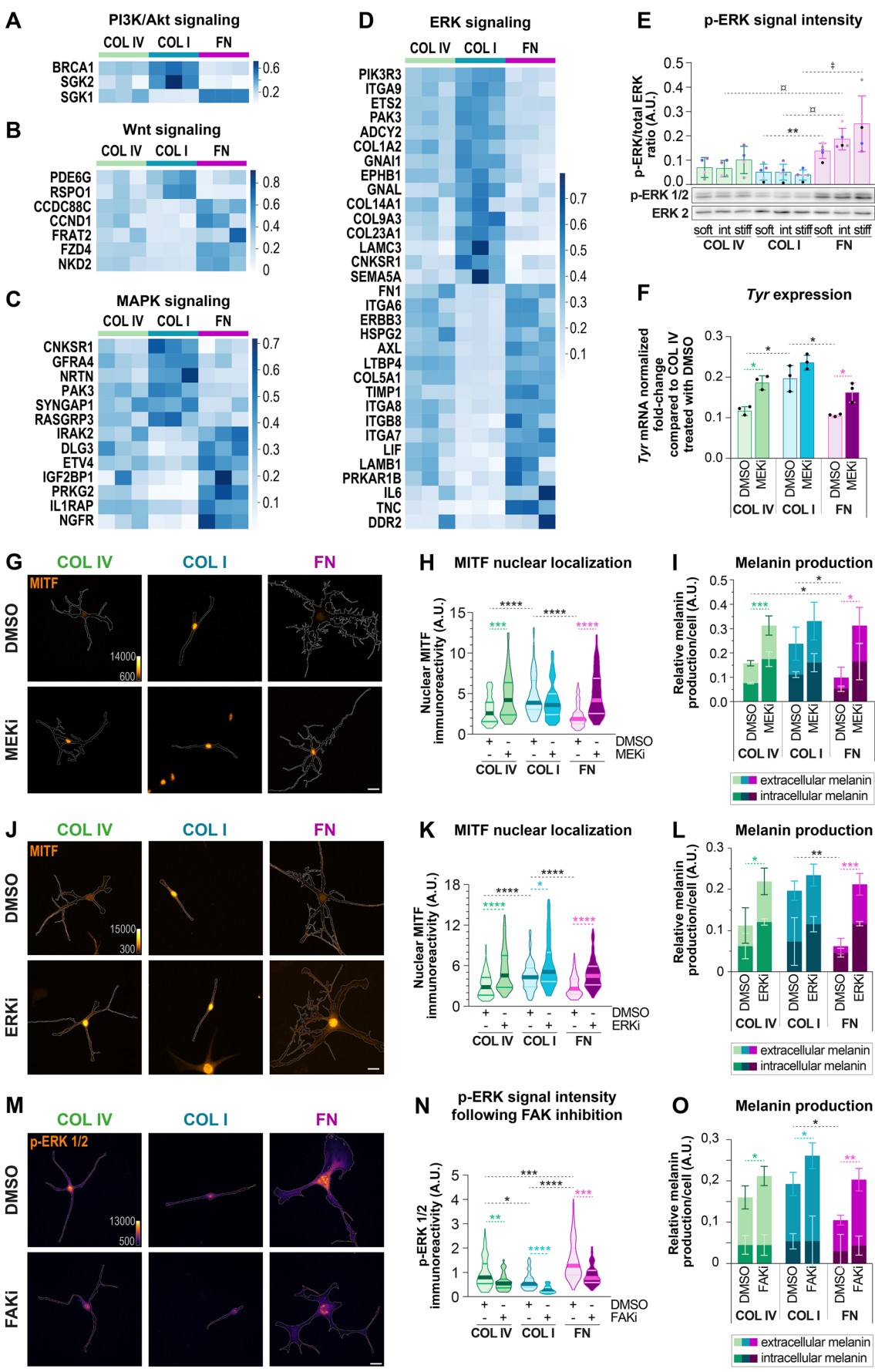

**Figure 6.  ECM-mediated ERK activation regulates MITF localization and melanogenesis.**

(A–D) iMCs were cultured overnight on stiff substrates coated with COL IV, COL I, or FN. Gene expression heatmaps of signaling pathway components across ECM conditions. Darker blue indicates higher transcript abundance; lighter blue indicates lower abundance; bold labels denote statistically differentially expressed genes. Heatmaps illustrate: (A) genes involved in the PI3K/Akt signaling; (B) components of the Wnt signaling; (C) genes associated with MAPK signaling; and (D) genes associated with ERK signaling. (E) iMCs were cultured overnight on substrates of varying stiffness (soft, intermediate or stiff) coated with COL IV, COL I or FN. Representative Western blot and associated graph depicting the quantification of pERK1/2 Thr202/Tyr204 to total ERK2 ratio; means ± SD; COL IV, $N = 4$ (biological replicates); COL I and FN, $N = 5$ (biological replicates); Welch ANOVA test: COL IV: ns $p = 0.9977$ (1:60 vs. 1:35), ns $p = 0.758$ (1:60 vs. 1:10), ns $p = 0.654$ (1:35 vs. 1:10); COL I: ns $p > 0.9999$ (1:60 vs. 1:35), $p = 0.8712$ (1:60 vs. 1:10), ns $p = 0.88$ (1:35 vs. 1:10); 1:60: **$p = 0.0066$ (COL I vs. FN), ns $p = 0.8299$ (COL I vs. COL IV), ns $p = 0.1007$ (FN vs. COL IV); 1:10: ‡$p = 0.0393$ (COL I vs. FN), ns $p = 0.2418$ (COL I vs. COL IV), ns $p = 0.1137$ (FN vs. COL IV); and Kruskal–Wallis test: FN: ns $p = 0.3116$ (1:60 vs. 1:35), ns $p = 0.0851$ (1:60 vs. 1:10), ns $p > 0.9999$ (1:35 vs. 1:10); 1:35: ¤ $p = 0.0195$ (COL I vs. FN), ns $p > 0.9999$ (COL I vs. COL IV), ¤ $p = 0.0485$ (FN vs. COL IV). (F–O) iMCs were cultured on stiff substrates coated with COL IV, COL I, or FN. 4 h post-plating, a 16-h (F–H) or 72-h (I) treatment with DMSO or 100 nM of MEK inhibitor (MEKi = Trametinib), a 16-h (J, K) or 72-h (L) treatment with DMSO or 10 µM of ERK inhibitor (ERKi = Ravoxertinib) or a 16-h (M, N) or 72-h (O) treatment with DMSO or 10 µM of FAK inhibitor (FAKi = Ifebemtinib) was commenced. (F) Quantification of *Tyr* mRNA expression after MEK inhibitor treatment; means ± SD, $N = 3$ (biological replicates); multiple *t*-tests: DMSO: *$p = 0,02948$ (COL IV vs. COL I), ns $p = 0.239076$ (COL IV vs. FN), *$p = 0.015087$ (COL I vs. FN); MEKi: *$p = 0.02948$ (COL IV vs. COL I), ns $p = 0.24224$ (COL IV vs. FN), *$p = 0.015087$ (COL I vs. FN); DMSO vs. MEKi: *$p = 0.012357$ (COL IV), ns $p = 0.139977$ (COL I), *$p = 0.029362$ (FN). (G) Representative immunostainings for MITF following MEK inhibition; scale bar: 10 µm. A pseudo-color intensity scale ("orange hot" color map) was applied to enhance visualization of MITF nuclear localization. (H) Quantification of (G); violin plots showing the medians and distributions of the integrated density of nuclear MITF per cell; $N = 3$ (biological replicates), $n$(cells) ≥63; Kruskal–Wallis test: DMSO: ****$p < 0.0001$ (COL IV vs. COL I), ns $p = 0.6551$ (COL IV vs. FN), ****$p < 0.0001$ (COL I vs. FN); DMSO vs. MEKi: ***$p = 0.0001$ (COL IV DMSO vs. COL IV MEKi), ns $p > 0.9999$ (COL I DMSO vs. COL I MEKi), ****$p < 0.0001$ (FN DMSO vs. FN MEKi). (I) Quantification of intra- and extracellular melanin content following MEK inhibition by spectrophotometry at 405 nm; means ± SD, $N = 3$ (biological replicates); multiple *t*-tests: DMSO: ns $p = 0.092307$ (COL IV vs. COL I), *$p = 0.019629$ (COL IV vs. FN), *$p = 0.019109$ (COL I vs. FN); DMSO vs. MEKi: ***$p = 0.000252$ (COL IV DMSO vs. COL IV MEKi), ns $p = 0.532253$ (COL I DMSO vs. COL I MEKi), *$p = 0.027828$ (FN DMSO vs. FN MEKi). (J) Representative immunostainings for MITF following ERK inhibition; scale bar: 20 µm. A pseudo-color intensity scale ("orange hot" color map) was applied to enhance visualization of MITF nuclear localization. (K) Quantification of (J); violin plots showing the medians and distributions of the integrated density of nuclear MITF per cell; $N = 3$ (biological replicates), $n$(cells) ≥89; Kruskal–Wallis test: DMSO: ****$p < 0.0001$ (COL IV vs. COL I), ns $p > 0.9999$ (COL IV vs. FN), ****$p < 0.0001$ (COL I vs. FN); DMSO vs. ERKi: ****$p < 0.0001$ (COL IV DMSO vs. COL IV ERKi), *$p = 0.0334$ (COL I DMSO vs. COL I ERKi), ****$p < 0.0001$ (FN DMSO vs. FN ERKi), ns $p > 0.9999$ (COL I DMSO vs. COL IV ERKi), ns $p > 0.9999$ (COL I DMSO vs. FN ERKi). (L) Quantification of intra- and extracellular melanin content following ERK inhibition by spectrophotometry at 405 nm; means ± SD, $N = 4$ (biological replicates); Welch ANOVA tests: DMSO: ns $p = 0.0693$ (COL IV vs. COL I), ns $p = 0.2556$ (COL IV vs. FN), **$p = 0.0057$ (COL I vs. FN); DMSO vs. ERKi: *$p = 0.0186$ (COL IV ERKi vs. COL IV DMSO), ns $p = 0.6035$ (COL IV ERKi vs. COL I DMSO), ns $p = 0.7619$ (FN ERKi vs. COL I DMSO), ***$p = 0.0004$ (FN ERKi vs. FN DMSO). (M) Representative immunostainings for p-ERK following FAK inhibition; scale bar: 20 µm. A pseudo-color intensity scale ("fire" color map) was applied to enhance visualization of p-ERK intensity. (N) Quantification of (M); violin plots showing the medians and distributions of the integrated density of p-ERK per cell; $N = 3$ (biological replicates), $n$(cells) ≥83; Kruskal–Wallis test: DMSO: *$p = 0,0483$ (COL IV vs. COL I), ***$p = 0,0008$ (COL IV vs. FN), ****$p < 0.0001$ (COL I vs. FN); DMSO vs. FAKi: **$p = 0.0021$ (COL IV DMSO vs. COL IV FAKi), ****$p < 0.0001$ (COL I DMSO vs. COL I FAKi), ***$p = 0.0004$ (FN DMSO vs. FN FAKi). (O) Quantification of intra- and extracellular melanin content following FAK inhibition by spectrophotometry at 405 nm; means ± SD, $N = 5$ (biological replicates); multiple *t*-tests: DMSO: ns $p = 0.258394$ (COL IV DMSO vs. COL I DMSO), ns $p = 0,058323$ (COL IV DMSO vs. FN DMSO), *$p = 0.005712$ (COL I DMSO vs. FN DMSO); DMSO vs. FAKi: *$p = 0.033822$ (COL IV DMSO vs. COL IV FAKi), *$p = 0.033196$ (COL I DMSO vs. COL I FAKi), **$p = 0.003493$ (FN DMSO vs. FN FAKi). AU arbitrary units. Source data are available online for this figure.

Together, these data demonstrate that FN-mediated MEK/ERK activation counteracts MITF nuclear accumulation and melanin synthesis, suggesting that MEK/ERK—potentially downstream of FAK— act as negative regulators of melanogenesis in response to ECM cues.

In summary, we deciphered that distinct ECM proteins in the MC environment steer melanogenesis through differential activation of the MEK/ERK pathway and subsequent regulation of MITF localization and activity. We further showed that COL I and FN exert opposite effects on MC behavior and functions: COL I elicits a highly pigmented and proliferative, but non-motile MC state, while FN triggers a less pigmented, low proliferative and highly motile MC state. Finally, by combining varying substrate stiffness and distinct ECM types, we delineated that stiffness-mediated MC functions rely on specific ECM components.

## Discussion

### ECM components fine-tune MC phenotypes and functions

Our study identifies ECM components as critical environmental triggers that instruct MC behavior. Through dynamic interactions with the ECM, MCs engage adhesion-dependent signaling, such as

FAK activation, enabling them to decode contextual ECM inputs and adapt their phenotype accordingly. Specifically, we observed that compared to the abundant physiological matrix protein that epidermal MCs face (COL IV), COL I, and FN– representative of a dermal matrix and thus an altered environment– elicit notable phenotypic shifts in MCs. Relative to COL IV, COL I reduced MC migration and increased melanin production, while FN promoted migration and decreased both proliferation and melanin production. These contrasting effects suggest that ECM composition can selectively modulate distinct aspects of MC behavior. Notably, the intermediate state observed on COL IV supports a model in which this basement membrane component enables MCs to maintain phenotypic flexibility—for example, allowing them to increase melanin production in response to external stimuli such as UV or inflammation. The opposing responses of MCs to FN and COL I underscore the importance of ECM composition in regulating MC function, suggesting that context-dependent ECM remodeling can actively shape MC behavior, with relevance not only for MC homeostasis but potentially also for pathophysiological states and stress responses.

Our findings open the possibility that ECM alterations can disrupt MC homeostasis, with potential beneficial or detrimental consequences for skin health depending on the context. For instance, wound healing reflects a complex physiological process in which MC phenotypic shifts could have a significant impact. Upon

skin injury, the initial stage of tissue repair entails the formation of a provisional FN-rich environment (Potekaev et al, 2021), which may facilitate the repopulation of MCs within the regenerated tissue (Snell, 1963). Taking into account our observation that FN enhances MC motility in vitro (Fig. 2), it is tempting to speculate that this FN-enriched tissue enables MCs to efficiently migrate into wound sites and re-establish their protective function, such as melanin-based protection from UV-induced damage. Conversely, in fibrotic conditions such as scleroderma, marked by stiffening of the skin due to excessive deposition of COL I, cases of localized hyperpigmentation have been reported in patients, possibly reflecting COL I-driven MC reprogramming. Vitiligo, which is characterized by the loss of epidermal MCs and non-pigmented patches, is associated with a decrease in COL IV and FN and a concomitant increase in COL I content (Rani et al, 2023). The repigmentation process requires MCs to migrate into depigmented areas and synthesize melanin effectively (Norris et al, 1994). Considering our observation of strongly reduced MC migration on COL I, it seems plausible that COL I enrichment in vitiligo lesions interferes with the efficient MC redistribution into depigmented areas in the skin. Furthermore, melasma is a multifactorial skin condition that is characterized by focal hypermelanosis. Sex hormones and UV radiation have been implicated in the development of melasma (Espósito et al, 2022). Interestingly, disruption of the basement membrane has also been reported in 96% of melasma lesions, with MCs protruding into the dermal layer in 66% of cases (Torres-Álvarez et al, 2011). While the mechanism of this dermal invasion by MCs remains open, it has been proposed that the migration of "hyperactive" MCs into the dermis leads to the constant hyperpigmentation in melasma (Phansuk et al, 2022; Torres-Álvarez et al, 2011). This concept would be in line with our findings of COL I-triggered melanogenesis, opening the possibility that MC hyperactivity in melasma could, at least in part, result from the exposure to dermal COL I.

Together, these examples illustrate how context-dependent ECM remodeling could shape MC behavior, with implications not only for physiological repair but also for pathological skin remodeling. Though further investigation is warranted, these observations open new avenues to explore how shifting ECM landscapes influence MC phenotypes in vivo.

## FN-induced phenotypic reprogramming rewires MCs toward a dedifferentiated state

Our study also highlights for the first time that ECM components are pivotal in regulating the phenotypic plasticity of MCs. We demonstrate that a FN-rich environment rewires MCs toward a dedifferentiated state, marked by reduced melanin production, slow-cycling, and increased motility, while COL I elicits features of a differentiated phenotype, promoting melanin synthesis but limiting migration. In addition to these phenotypic observations, our data indicate that FN exposure induces a distinct transcriptional program associated with MC dedifferentiation. Transcriptomic profiling revealed that FN-cultured MCs downregulate melanocytic differentiation markers and upregulate genes linked to plasticity, stemness, and neural crest-like features. This phenotype is indicative of an adaptive, dedifferentiated state, distinct from the more stable melanogenic profile observed on both

COL I and COL IV, and suggests that FN may act as a cue for reprogramming MC identity.

Notably, this dedifferentiated signature observed in FN-exposed MCs is reminiscent of the early phenotypic changes reported during malignant transformation of MCs. Indeed, in a mouse model recapitulating features of human melanomagenesis, it has been shown that mature MCs expressing a B-Raf oncogene can undergo transcriptional reprogramming, associated with a loss of their differentiated characteristics and eventual invasion into the dermis (Köhler et al, 2017). This suggests that MC dedifferentiation precedes MC transformation and melanoma development. Hence, the dedifferentiated phenotype observed in FN-exposed MCs could reflect one of several steps that are required during early stages of melanoma initiation. Taken together, this raises the possibility that an FN-rich environment, combined with oncogenic mutations, could render MCs susceptible to transformation and ultimately melanomagenesis.

In conclusion, our findings underscore the remarkable plasticity of MCs and their ability to shift differentiation states in response to environmental cues, highlighting the importance of ECM molecules in regulating MC behavior. By modulating the balance between melanogenic identity and motile potential, the ECM—particularly FN—may influence not only physiological processes like tissue repair but also pathological trajectories, including pigmentation disorders and oncogenesis. Understanding how different conditions elicit these shifts could provide valuable insights into normal skin physiology, the mechanisms underlying melanocyte-related conditions like melanoma, and potential therapeutic targets for restoring normal melanocyte function in pigmentation disorders. In particular, elucidating how ECM remodeling contributes to MC dedifferentiation could open new therapeutic avenues in both regenerative medicine and cancer biology.

## ECM-dependent ERK activation controls MITF localization and activity, and melanogenesis

Given the well-established role of MITF as a master regulator of MC differentiation, pigmentation, and survival, and its known regulation by ERK signaling, we reasoned that it could function as a key integrator of ECM-derived cues. We therefore centered our analysis on MITF, aiming to understand how its localization and activity might be shaped by the extracellular microenvironment. This hypothesis was supported by our finding that ECM-dependent phenotypic shifts of MCs are tightly linked to differential activation of the MEK/ERK pathway, which in turn governs the localization and output of MITF. Specifically, COL I limits ERK activity, triggering elevated nuclear MITF, while FN stimulates high ERK activity, resulting in reduced nuclear MITF levels. This is further associated with the adaptation of MC functions under the control of MITF, whereby proliferation and melanin production are enhanced on COL I but reduced on FN. In line with previous studies that linked RAF and MEK/ERK activation to cytoplasmic retention and nuclear export of MITF in melanoma cells, respectively (Estrada et al, 2022; Ngeow et al, 2018), we here demonstrate in MCs that MEK and ERK inhibition reverted the low nuclear MITF and melanin production in FN-exposed MCs. This, together with our findings on ECM-dependent differential ERK activation in MCs, identifies a hitherto unrecognized role of ECM cues in MITF nuclear localization through the regulation of ERK

activity. Our data further suggest that FAK may act upstream of ERK, as FAK is more activated on FN, and its inhibition on FN reduced ERK activation and melanin production.

Our transcriptomic analysis also pointed to ECM-induced expression changes in Wnt pathway components, e.g., an upregulation of *Frat2* and *Ccnd1* on FN. FRAT2 is known to stabilize β-catenin by inhibiting its GSK3β-mediated degradation (van Amerongen and Berns, 2005), thereby supporting β-catenin–dependent transcription. Since β-catenin can enhance MITF transcription through TCF/LEF-mediated activity (Widlund et al, 2002; Yasumoto et al, 2002), and given that MITF itself can interact with β-catenin to regulate gene expression (Schepsky et al, 2006), these findings open the possibility that Wnt signaling, next to FAK/MEK/ERK signaling, could contribute to ECM-dependent modulation of MITF levels and function.

Overall, further studies are needed to dissect the specific molecular signals downstream of FN that drive the shift of mature MCs toward less differentiated states and to delineate the respective contributions of individual signaling pathways in ECM-mediated responses.

## Interplay between ECM components and substrate stiffness in the mechanosensation of MCs

Our findings underpin a crucial role for ECM components in regulating mechanosensation in MCs and show that ECM components and substrate stiffness govern MC functions. So far, studies have predominantly explored either the roles of different ECM subtypes using ultra-hard substrates like glass or plastic (Hara et al, 1994), or focused on the impact of substrate stiffness with a single ECM type (Choi et al, 2014). In contrast, our approach, combining varying substrate stiffness with different ECM proteins, demonstrated that (1) over the range of stiffness and ECM molecules tested, matrix protein subtypes can dominate over mechanical substrate surface properties, as altering ECM proteins produced more pronounced effects on MC behavior compared to changes in substrate stiffness; and (2) the ability of MCs to respond to mechanical cues is highly dependent on the specific ECM proteins present. Notably, COL I supports stiffness-induced MITF nuclear localization and MC proliferation, while FN enhances mechanosensitive responses at the level of FAK activation and number of FAs. Seong et al previously reported that in a human fibrosarcoma cell line, stiffness-mediated FAK activation can be observed on FN- but not on COL I-coated substrates (Seong et al, 2013). While our observations in MCs are consistent with such effects on FN, iMCs cultured on COL I even showed reduced FAK activity with substrate stiffness, and pMCs had the lowest FAK activity at intermediate stiffness. These observations suggest cell-type-specific roles of ECM molecules in mechanoresponses.

Strikingly, our data identified MITF as a novel ECM- and mechanosensitive transcription factor, with a stiffness-dependent increase as well as the highest nuclear localization observed on COL I compared to COL IV and FN. Such fine-tuning of MITF subcellular localization may ensure that MC responses are appropriately matched to the environmental matrix protein composition and mechanical features of the ECM. Overall, this highlights the context-dependent nature of MITF regulation downstream of ERK, revealing a hitherto unknown aspect of MITF modulation by ECM cues. Furthermore, our findings suggest that

ECM components and substrate stiffness do not merely exert additive effects; instead, individual ECM types demonstrate synergistic interactions with stiffness in regulating MC behavior. This underscores the importance of considering both ECM protein composition and mechanical properties in future studies, as the interplay between these factors significantly influences how cells interpret and respond to mechanical signals, thereby affecting cellular function and differentiation pathways.

## Conclusion

In summary, we report a novel ECM-dependent FAK/MEK/ERK/MITF axis that controls the plasticity of MCs in response to interactions with their environment. Our findings underscore the critical role of the ECM in orchestrating MC function and differentiation, highlighting the need for further studies to fully understand the intricate crosstalk between mechanical parameters and ECM ligands in the cellular microenvironment (Fig. 7). Moreover, our data point to a complex link between ECM protein types and substrate stiffness, which can influence MC mechanosensation (Fig. 7). Elucidating the specific molecular mechanisms driving this interplay, particularly the roles of MITF and other key regulators, will be crucial for a comprehensive understanding of MC function and pathology.

## Methods

### Reagents and tools table

| Reagent/resource | Reference or source | Identifier or catalog number |
|---|---|---|
| **Experimental models** | | |
| C57Bl6/N (*M. Musculus*) | Ibis cells, Iden lab | |
| C57Bl6/N (*M. Musculus*) | Primary melanocytes, Iden lab | |
| **Antibodies** | | |
| Mouse anti-MITF antibody [C5] | Abcam | Ab12039 |
| Mouse anti-BrdU | Biolegend | 339802 |
| Mouse anti-Ki67 | Biolegend | 652401 |
| Mouse anti-ERK2 | BD Pharmingen | 610104 |
| Rabbit anti-phospho-FAK antibody (Tyr397) | Cell Signaling Technology | 3283S |
| Rabbit anti-YAP (D8H1X) XP® | Cell Signaling Technology | 14074S |
| Rabbit anit-pERK1/2 | Cell Signaling Technology | 4370 |
| Rabbit anti-MITF | Cell Signaling Technology | 97800 |
| Rabbit anti-Calnexin | Enzo Life Sciences | ADI-SPA-860-D |
| Alexa Fluor™ 488 donkey anti-rabbit | Invitrogen | A-21206 |
| Alexa Fluor™ 647 goat anti-mouse | Invitrogen | A-21235 |
| HRP donkey anti-rabbit | Invitrogen | A16023 |

| Reagent/resource | Reference or source | Identifier or catalog number |
|---|---|---|
| HRP sheep anti-mouse | Amersham | NA931V |
| Alexa Fluor™ 568 Phalloidin | Molecular Probes | A12380 |
| DAPI | Carl Roth | 6335.1 |
| **Oligonucleotides and other sequence-based reagents** | | |
| GAPDH mouse Mm99999915_g1 | Thermo Fisher Scientific | 4448489 |
| Tyr mouse Mm00495817_m1 | Thermo Fisher Scientific | 4331182 |
| ON-TARGETplus mouse Mitf siRNA | Dharmacon | L-047441-00-0005 |
| siGENOME control siRNA | Dharmacon | D-001206-14-20 |
| **Chemicals, Enzymes and other reagents** | | |
| Bovine plasma fibronectin | Sigma-Aldrich | F1141 |
| Bovine serum albumin | Sigma-Aldrich | A7906 |
| BrdU | Roche | #10280879001 |
| Cholera toxin | Sigma-Aldrich | #C8052 |
| Collagen G type I from calf skin | Sigma-Aldrich | L7213 |
| Cultrex™ Mouse Collagen IV | Bio-Techne | #3410-010-02 |
| D-PBS | Gibco | 14190094 |
| Dispase® II | Sigma-Aldrich | #D4693 |
| DMSO | Fisher Bioreagents | BP231-100 |
| EDTA | Sigma-Aldrich | E1644 |
| Fetal calf serum | Sigma-Aldrich | #S0615 |
| Fish gelatin | Sigma-Aldrich | #G7765 |
| Fluoromount-G™ | Thermo Fisher Scientific | #00-4958-02 |
| Glycine | Carl Roth | 3187.5 |
| HEPES | Carl Roth | #9105.4 |
| Ifebemtinib | Selleckchem | #E11144 |
| Milk powder | Carl Roth | #T145.2 |
| NaCl | Carl Roth | #3957.1 |
| NaOH | Carl Roth | 6885.2 |
| Non-essential amino acids | Gibco | 11140-035 |
| Paraformaldehyde | Carl Roth | 0335.1 |
| Penicillin/Streptomycin | Gibco | 15140-122 |
| Ravoxertinib | Selleckchem | #S7554 |
| RPMI medium | Gibco | 61870-010 |
| RPMI medium, phenol red-free | Gibco | 11835036 |
| SDS | Carl Roth | 0183.5 |
| Sodium pyruvate | Gibco | 11360-039 |
| TPA | Sigma-Aldrich | #P8139 |
| Trametinib | Biozol | #10999 |
| Triton X-100 | Fisher Bioreagents | BP151-100 |
| TRIzol reagent | Thermo Fisher Scientific | Invitrogen#15596026 |
| TrypLE select | Gibco | #12563-011 |
| Trypsin/EDTA | Gibco | 25300-054 |

| Reagent/resource | Reference or source | Identifier or catalog number |
|---|---|---|
| Tween-20 | Carl Roth | #9127.1 |
| Western Lightning® Plus-ECL | PerkinElmer | NEL104001EA |
| iTaq Universal Probes Supermix | Bio-Rad | 1725131 |
| QuantiTect Reverse Transcription Kit | Qiagen | 205311 |
| SYLGARD 184 Elastomer Kit | Dow Silicones | 1003993081 |
| Thermo Scientific™ Pierce™ BCA Protein Assay Kit | Thermo Scientific | 10678484 |
| Viromer® Blue kit | Lipocalyx | VB-01LB-03 |
| RNeasy Micro Kit | Qiagen | 217084 |
| **Software** | | |
| Amersham ImageQuant™ 800 control software | Cytiva | |
| CFX Opus 96 | Bio-Rad | |
| GraphPad PRISM, v.10.2.1 | GraphPad Software | |
| ImageJ/ Fiji | Open-source software | |
| JPK SPM Data Processing Software v8.0.1 | Brunker, Nano | |
| ZEN Blue 2.6 | Zeiss | |
| **Other** | | |
| Axiocam 305 camera | Zeiss | |
| AxioObserver Z1 inverted microscope | Zeiss | |
| Biosphere B2000 | Nanotools | |
| Colibri 7 LED light source | Zeiss | |
| EC Plan Neofluar 40x/1.3 oil objective | Zeiss | |
| ibiTreat μ-Dish | Ibidi | 81156 |
| Multiskan™ FC Microplate Photometer | Thermo Fisher Scientific | |
| NanoDrop One© spectrophotometer | Thermo Fisher | |
| Nanowizard 4 | Bruker Nano GmbH | |
| PDC-002-CE plasma cleaner | Harrick Plasma | |
| Plan-Apochromat 20x/0.8 air objective | Zeiss | |
| Plan-Apochromat 63x/1.4 oil objective | Zeiss | |
| Rolera ECM2 camera | Zeiss | |
| Zeiss CellDiscoverer 7 | Zeiss | |
| μ-Slide 8 well | Ibidi | 80841-90 |

## Methods and protocols

### Preparation of PDMS substrates with tunable stiffness and ECM protein coating

Polydimethylsiloxane (PDMS) substrates were prepared using a SYLGARD 184 Elastomer kit (Dow Silicones, USA) and varying the

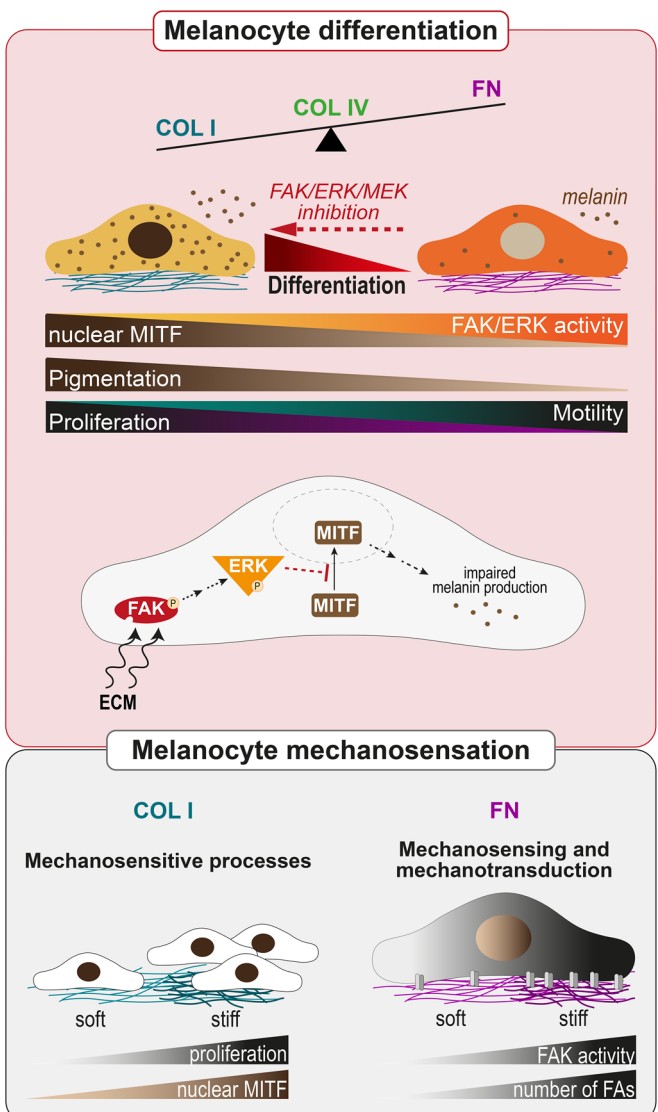

**Figure 7.  (graphical abstract).**

Upper panel: MC differentiation is differentially modulated by ECM components. COL I promotes a differentiated phenotype by limiting ERK activation, leading to high nuclear MITF levels, associated with increased pigmentation and proliferation but reduced motility. In contrast, FN rewires MCs towards dedifferentiation, characterized by enhanced ERK activity resulting in reduced nuclear MITF and decreased pigmentation, and associated with a slow-cycling phenotype as well as increased motility. Reducing ERK activity, through FAK, MEK or ERK inhibition, restores the differentiation of MCs exposed to FN. Lower panel: ECM components influence how cells interpret and respond to mechanical signals: COL I enables stiffness-mediated MITF nuclear localization and MC proliferation, while FN supports a stiffness-dependent increase in focal adhesion number and FAK activation.

crosslinker:silicone base agent ratio to 1:60, 1:35, and 1:10, respectively (Fig. EV1). Once mixed, the PDMS elastomers were cured at 60 °C for 16 h. To enhance the hydrophilicity of the substrates and facilitate binding of ECM proteins, all substrates underwent argon plasma treatment using a PDC-002-CE plasma cleaner (Harrick Plasma, USA) at 300 mTorr for 2 min. The substrates were then coated for 2 h at 37 °C with 30 μg/ml of mouse collagen IV (Cultrex™, #3410-010-02,

Bio-Techne, Germany), 30 μg/ml of type I collagen from calf skin (Collagen G, L7213, Sigma-Aldrich, USA), or 10 μg/ml of bovine plasma fibronectin (F1141, Sigma-Aldrich, USA) diluted in sterile D-PBS (Gibco, USA). After incubation, the excess ECM solution was aspirated, and substrates were washed with sterile D-PBS prior to cell seeding.

### Determination of the elastic modulus of uncoated and coated PDMS surfaces

Atomic force microscopy (AFM) served to determine the elastic modulus (Young's Modulus) of uncoated and protein-coated PDMS surfaces used in this study. For this, the PDMS elastomers (ratio 1:10, 1:35 or 1:60) were prepared in an AFM-suitable imaging dish (ibiTreat μ-Dish, Ibidi, Germany) and transferred to the sample stage of the microscope (Nanowizard 4, Bruker Nano GmbH, Germany). An AFM cantilever with a spherical tip (Biosphere B2000, Nanotools, Germany) with a nominal spring constant of 0.2 N/m and a sphere radius of 2 μm was used to perform nanoindentation measurements. AFM force-distance measurements were conducted in a grid of $4 \times 4$ points with a lateral distance of 4 μm (=one tip diameter) between two points to probe native spots. The AFM cantilever was extended to the PDMS surface at a speed of 1 μm/s and pressed on it with a loading force of 20 nN. Afterwards, the cantilever was retracted with 1 μm/s to complete the force-distance measurements. For each force-distance curve, the Hertz fit (Horvath et al, 2019) was applied to the contact region of the extended curve to evaluate the Young's Modulus with the JPK SPM Data Processing Software v8.0.1 (Bruker Nano GmbH, Germany).

### Mice

All mice were housed and maintained according to federal guidelines within the specific pathogen-free (SPF) animal facility in Building 61.4 of the Saarland University in Homburg under the breeding license of Sandra Iden's lab (§11 Az. 2.4.1.1-Iden). All experiments were performed in line with German animal welfare laws as well as institutional guidelines and were reported to the responsible authorities (Vorabmeldung VM2023-16 and previous notifications). Wildtype (wt C57Bl6/N) mice were either purchased from the Jackson Laboratory (Charles River Germany) or were taken from in-house breedings of transgenic mouse lines. In those cases, F1 and F2 generation were genotyped to verify the wt alleles of the genes of interest of the respective mouse line.

### Isolation and culture of primary mouse melanocytes

Primary mouse MCs were isolated from the epidermis of newborn wt mice (C57Bl6/N, male and female) and maintained up to passage 3 as previously described (Mescher et al, 2017). Briefly, to separate epidermis from dermis, whole skin from mice at postnatal day (P)0-P3 were incubated in a solution of 5 mg/ml Dispase II (#D4693, Sigma-Aldrich, USA) diluted in RPMI medium (Gibco, USA) supplemented with 10% fetal calf serum (FCS, #S0615, Sigma-Aldrich, USA), penicillin (100 U/ml), streptomycin (100 μg/ml), 100 nM sodium pyruvate, and 10 mM non-essential amino acids (all Gibco, USA) at 4 °C overnight. The next day, the epidermis was incubated for 20 min in TrypLE select (#12563-011, Gibco, USA) at room temperature. Dissociated cells were collected and cultured in RPMI medium containing 200 nM TPA (12-O-tetradecanoylphorbol-13-acetate,

#P8139, Sigma-Aldrich, USA) and 200 pM cholera toxin (#C8052, Sigma-Aldrich, USA), hereafter referred to as RPMI+. At passage 0, the cell cultures contained non-pigmented melanoblasts, MCs, and keratinocytes. After 7 days, cells were passaged, and due to terminal differentiation, keratinocytes were largely removed, resulting in MC monocultures suitable for further experiments.

### Generation of stable murine melanocyte cell lines (iMCs)

Immortalized MCs (iMCs) were generated by immortalizing mouse primary MCs (mTmG-positive) with SV40 large T antigen, followed by subcloning. The cells express membrane-targeted tandem dimer Tomato (dt; Muzumdar et al, 2007), yielding red fluorescence. iMCs exhibit morphological and molecular characteristics common to normal murine skin MCs. iMCs were maintained in culture in RPMI+ at 37 °C with 5% $CO_2$. iMCs were tested for mycoplasma contamination using a PCR-based assay, and confirmed negative prior to experiments.

### Immunofluorescence

To assess cell area and morphology, and for quantification of p-FAK, Ki67, MITF, YAP, and p-ERK signal intensities, MCs were seeded on the different substrate combinations at a density of 2500 cells/cm², to avoid overcrowding and allow single cell measurements. Twenty-four hours later, MCs were fixed in 4% PFA/PBS for 10 min. For MITF, YAP, and p-FAK stainings, cells were blocked for 1 h at room temperature with 5% bovine serum albumin (BSA) in TBS containing 0.2% Triton X-100 (named blocking buffer hereafter). Primary antibody incubation was performed in blocking buffer diluted at a ratio of 1:5 in TBS at 4 °C overnight. For Ki67 and p-ERK stainings, blocking and primary antibody dilution were performed using PB buffer composed of 0.05% milk powder (#T145.2, Carl Roth, Germany), 0.25% fish gelatin (#G7765, Sigma-Aldrich, USA), 0.5% Triton X-100 (#BP151-100, Fisher Bioreagents, Germany), 20 mM HEPES (pH 7.2, #9105.4, Carl Roth, Germany), and 0.9% NaCl (#3957.1, Carl Roth, Germany). For all, after 3x washing in 0.2% Tween/TBS, MCs were stained with AlexaFluor-conjugated secondary antibodies, DAPI (Carl Roth, Germany) and phalloidin to stain actin for pMCs for 1 h at room temperature. After washing, PDMS scaffolds were mounted onto glass coverslips using Fluoromount-G™ (#00-4958-02, Thermo Fisher Scientific, USA). Antibodies used for immunostainings are listed in the Antibodies and reagents section.

### BrdU assay

Forty-eight hours post-plating, MCs were incubated with 160 µg/ml of BrdU (5-Brom-2′-desoxyuridin; #10280879001, Roche, Germany) for 2 h at 37 °C and then washed with PBS and fixed with 4% PFA for 10 min at room temperature. Cells were treated with 2 N HCl for 10 min at room temperature to achieve DNA denaturation, followed by a PBS wash and subsequent immunofluorescence staining as previously described. BrdU-positive MCs were visualized using fluorescence microscopy (see below) and quantified from at least 20 randomly positioned micrographs. The background signal was subtracted from the measured signal of each nucleus to identify BrdU-positive cells. The proliferation rate was then determined by counting BrdU-positive cells and dividing this number by the total number of cells, identified via DAPI staining.

### Microscopy

Epifluorescence micrographs were acquired with an AxioObserver Z1 inverted microscope equipped with a Colibri 7 LED light source (Zeiss, Germany) using a Plan-Apochromat 20x/0.8 air objective, an EC Plan Neofluar 40x/1.3 oil objective, a Plan-Apochromat 63x/1.4 oil objective and an Axiocam 305 camera (Zeiss, Germany) or Rolera ECM2 camera coupled to the ZEN Blue imaging software V2.6 (Zeiss, Germany). To minimize selection bias, micrographs were acquired in a blinded and unbiased manner: the operator acquired images of the first cell encountered in each field of view without pre-selection based on cell morphology or signal intensity.

### Image quantification

Image analysis was performed using Fiji/ImageJ software (Schindelin et al, 2012; Schneider et al, 2012) to quantify cell area, morphology, FAK phosphorylation and focal adhesion numbers, ERK phosphorylation, as well as YAP and MITF nuclear translocation in MCs 24 h post-plating.

MC dendricity and cell area.  Epifluorescence micrographs of actin (phalloidin staining) or td-Tomato, for pMCs and iMCs, respectively, were preprocessed using a "mean" filter to smoothen images, by reducing noise and averaging pixel values, followed by the "median" filter, which further reduced noise while preserving edges and details. Images were then converted into binary images by using the "Threshold" function to obtain a mask of the cell shape. The binary mask was used either for cell area quantification or for the quantitative assessment of morphological characteristics of MCs. Using the "Sholl Analysis" plugin, concentric circles (named radius), were drawn on the binary image at regular intervals (5 µm) from the center of the cell body. The number of intersections between dendrites and each radius was counted and then plotted to create a Sholl profile, which shows the number of dendrite intersections at increasing distances from the cell body (Binley et al, 2014; Sholl, 1953).

p-FAK immunoreactivity.  To quantify p-FAK immunoreactivity in MCs, actin (phalloidin staining) or td-Tomato signals were used to detect the cell contour of pMCs and iMCs, respectively. This contour was then superimposed on the p-FAK signal, followed by measurement of the integrated density of the signal of individual MCs. For iMCs, data were pooled after normalization for each experiment, as two different microscope cameras have been used for these experimental series.

Focal adhesion number.  To quantify the number of focal adhesions based on p-FAK signals, first, a background subtraction was performed using a rolling ball algorithm. Then, a suitable threshold was applied to isolate the focal adhesion signal from the background. Individual focal adhesions were detected using the "analyze particles" function (0.05–5 µm). Finally, the number of focal adhesions per cell and per unit area was determined.

YAP and MITF subcellular localization.  Immunodetection of MITF and YAP were performed in separate experiments and analyzed independently. For both stainings, appropriate negative controls were included (secondary antibody alone without primary antibody), which showed no detectable signal.

To evaluate YAP nuclear localization in response to ECM cues, DAPI was used to delineate nuclear contours, which were overlaid

onto the YAP signal to quantify nuclear integrated intensity in individual MC. To account for signal variation across experiments and due to the use of two different cameras along this experimental series, values were normalized within each independent experiment by dividing each individual measurement by the mean nuclear YAP intensity across all conditions.

For MITF quantification, phalloidin-based F-actin staining (in case of pMCs) or td-Tomato signal (for iMCs) was employed to delineate whole-cell contours, and DAPI served as nuclear counterstain. These masks were overlaid onto the MITF signal to extract integrated intensity values from the nuclear and whole-cell compartments of individual MCs. Cytoplasmic MITF intensity was calculated by subtracting the nuclear signal from the whole-cell signal. Then, nuclear and cytoplasmic intensity values were normalized within each independent experiment by dividing the values of each individual measurement by the mean total intensity across all conditions, thereby normalizing for inter-experimental variability while preserving relative differences between ECM conditions. Finally, MITF nuclear localization was expressed as the percentage of nuclear signal relative to total cellular signal (value for immunoreactivity in the nucleus/value for immunoreactivity in the whole cell × 100), allowing comparison of nuclear enrichment across ECM conditions.

p-ERK immunoreactivity.  To quantify p-ERK immunoreactivity in iMCs, the td-Tomato signal was used to detect the cell contour, which was then superimposed on the p-ERK immunostaining signal, followed by measurement of the integrated density of the signal of individual iMCs. Values were normalized within each independent experiment by dividing the values of each individual measurement by the mean nuclear intensity across all conditions. This approach allowed us to account for variability between experiments while preserving relative differences between ECM conditions.

### Live-cell imaging for analysis of MC motility

Live-cell imaging was conducted to investigate dynamic cellular behaviors in response to the ECM. Taking advantage of the stable expression of the fluorescent membrane-targeted Tomato, MCs were seeded at a density of 1000 cells/cm² in a plastic chamber slide coated with the distinct ECM proteins (μ-Slide 8 Well, Ibidi, Germany), and placed in a live-cell imaging system (Zeiss CellDiscoverer 7) equipped with a temperature- and $CO_2$-controlled chamber. Time-lapse images were acquired every 20 min for 20 h using the epifluorescence microscopy mode. Two methodologies were employed to enable real-time dynamic visualization and quantitative analysis of cell migration. To quantify migration parameters, i.e., total distance migrated by cells, and their speed, cells were tracked using the "Manual tracking" plugin in Fiji software. Additionally, the "Temporal Color Code" in Fiji software was applied to track cell shape and movement over time, allowing to visualize changes of MC dynamics over time. Briefly, different colors represent the shape and position of cells at each time point, with the color gradient indicating the progression of time. The overlay of different colors provides information on the dynamic behavior of cells throughout the observation period.

### RT-qPCR

For quantitative RT-PCRs, RNA was isolated from MCs using TRIzol reagent according to the manufacturer's protocol (Invitrogen#15596026, Thermo Fisher Scientific, USA). RNA was reverse transcribed using a QuantiTect Reverse Transcription kit (Qiagen,

Germany) and amplified with iTaq Universal Probes Supermix (Bio-Rad, USA). Target gene expression was detected using TaqMan probes (GAPDH mouse Mm99999915_g1; Tyr mouse Mm00495817_m1; Thermo Fisher Scientific, USA). Gene expression changes were calculated using the comparative CT (ΔΔCT) method, normalized to GAPDH expression, and compared to cell lysates from soft substrates coated with COL IV.

### Melanin content assay

The melanin assay was performed to quantify the average melanin amount produced by a cell. Both intracellular melanin and extracellular melanin content (i.e., melanin released into the medium) were measured from the same batch of cells. For melanin assays, each condition was performed in duplicate: one well was used for melanin extraction, and the other for cell counting. MCs were cultured in 12-well plates for 72 h in 1 mL phenol red-free RPMI medium (Gibco, USA) to avoid interference with extracellular melanin measurements. The medium was then collected and analyzed to measure extracellular melanin content. Intracellular melanin was quantified following an adapted protocol from Ito and Wakamatsu (Ito and Wakamatsu, 2003). MCs were lysed in 400 μL of 10% DMSO in 1 N NaOH (extraction buffer) and incubated with agitation (24 h at 100 °C for pMCs and 18 h at 80 °C for iMCs). Extracellular extracts were quickly centrifuged for 10 s to sediment cellular debris, and intracellular extracts were centrifuged for 5 min at 13.000 rpm. Extra- and intracellular extracts and their respective blanks, RPMI media or lysis buffer, were loaded onto a 96-well plate and absorbance was measured at 405 nm using a Multiskan™ FC Microplate Photometer (Thermo Fisher Scientific, USA). For analysis, the OD values were subtracted from the blank to remove background signals caused by the medium or lysis buffer. Next, extracellular values were multiplied by 5 and intracellular by 2 to account for the stock volume, and the values were divided by the cell number to obtain the relative melanin content per cell.

For intracellular melanin quantification following MITF depletion, iMCs were transfected with 50 nM siMITF (ON-TARGETplus Mouse Mitf siRNA SMARTPool, L-047441-00-0005, Dharmacon, UK) or control siRNA (siGENOME Control Pool, D-001206-14-20, Dharmacon, UK) using the Viromer® Blue kit (Lipocalyx) 4 h after plating. Cells were harvested and lysed for melanin extraction 72 h post-transfection using the same procedure as above.

### Transcriptomic analysis

RNA extraction, quality control, library preparation, and sequencing. Total RNA was extracted using TRIzol reagent (Invitrogen, Thermo Fisher Scientific, USA) or the RNeasy Mini Kit (Qiagen, Germany), following the manufacturer's instructions. For each condition, RNA was prepared from three biological replicates. RNA quantity and purity were assessed using a NanoDrop One© spectrophotometer (Thermo Fisher Scientific, USA), and RNA integrity was verified using the RNA integrity number (RIN). Only samples with RIN values ≥4 were processed further. As additional quality controls, melanin content assays and RT-qPCR were performed in parallel, targeting melanocyte-specific markers *Mitf* and *Tyr*.

RNA sequencing was performed by Novogene (Munich, Germany) using standard Illumina protocols. Polyadenylated mRNAs were enriched, fragmented, and reverse-transcribed into cDNA. Strand-specific libraries were prepared and quality-checked using Qubit, real-time PCR, and a Bioanalyzer, then sequenced on

Illumina platforms (2 × 150 bp), generating ~25 million paired-end reads per sample. Raw reads were processed with fastp to remove adapters and low-quality reads. Clean reads were aligned to the mouse genome (GRCm39/mm39) using HISAT2, and gene expression was quantified with featureCounts and normalized as FPKM values. RNAseq data have been deposited at GEO (accession GSE297747).

Bioinformatics analysis. Gene expression data from cells cultured on COL1, COL4, and FN were processed using Pandas (McKinney, 2010) and NumPy (Harris et al, 2020) for sorting and initial exploration. Differential gene expression analysis was performed using the Bioconductor package DESeq2 (Love et al, 2014) to identify genes significantly regulated between the conditions. Genes with an adjusted p-value < 0.05 and |log2 fold change|≥1 were considered differentially expressed. Gene expression values were initially normalized by dividing the raw counts by the total number of genes in each sample, and then further normalized using the Euclidean method. Heatmaps were created using Seaborn (Waskom, 2021) to visualize expression patterns across conditions. Results were visualized using volcano plots generated with Matplotlib (Hunter, 2007). To explore biological relevance, the association of significantly regulated genes with biochemical pathways was analyzed using Reactome Pathway (Milacic et al, 2024), KEGG Pathway (Kanehisa et al, 2025), and WikiPathways (Agrawal et al, 2024). In addition, Gene Ontology (GO) (Thomas et al, 2022) enrichment analysis was performed to identify overrepresented biological processes, providing insight into the functional impact of ECM proteins on gene regulation.

### Preparation of cell extracts, SDS-PAGE, and immunoblotting
MCs cultures were lysed using boiled SDS/EDTA solution (1% SDS, 10 mM EDTA) and genomic DNA was sheared by passing the lysates through a 27 G x ¾ canula. Protein concentrations were quantified using the BCA assay according to the manufacturer's protocol (Pierce, Thermo Fisher Scientific, USA). SDS-PAGE was carried out using 8-10% polyacrylamide gels. After completing gel electrophoresis, the proteins were transferred onto a PVDF membrane at 20 V for 60 min. Following the transfer, the membrane was blocked for 1 h at room temperature using 5% BSA/TBST (TBS containing 0.2% Tween-20). The primary antibody was incubated in 5% BSA/TBST overnight at 4 °C on a roller. Membranes were then washed 3x for 10 min with TBST and incubated with an HRP-conjugated secondary antibody in 5% BSA/TBST for 1 h at room temperature on a roller. Membranes were washed 3x for 10 min with TBST, excess TBST was removed, and membranes were incubated 1 min with in a 1:1 mixture of ECL solutions A and B (PerkinElmer, USA). Excess ECL solution was removed, and the membrane was imaged using the Amersham ImageQuant™ 800 detector (Cytiva, USA). Following a 30-min wash at room temperature with a buffer containing 0.2 M glycine and 0.1% SDS (pH 2.5) to strip the membranes, they were blocked and immunoblotted for total ERK using the same membranes previously used for phospho-ERK detection.

### Quantification of phospho-ERK signal in immunoblot analyses
The protein band intensity of non-saturated Western blot signals was quantified using Fiji software. Each band was selected from the original. tiff image obtained with the Amersham ImageQuant™ 800 Detector, and the "Plot Lanes" tool generated intensity profiles. Peaks corresponding to individual bands (related to total or phosphorylated protein) were identified, and the area under them was measured. The p-ERK/total ERK ratio was calculated to assess relative ERK activity under the different conditions. The values were normalized to the sum of each membrane's values to determine the mean and standard deviation across experiments.

### Inhibitor treatment
To assess the role of ERK activity for ECM-dependent MC functions, MCs were seeded 4 h prior a 24-h- or 72-h-treatment with 100 nM Trametinib (MEK inhibitor, #10999, Biozol, Germany), 10 µM of Ravoxertinib (ERK inhibitor, #S7554, Selleckchem, Germany) or 10 µM of Ifebemtinib (FAK inhibitor, #E1114, Selleckchem, Germany) diluted in RPMI + .

### Antibodies
Details on antibodies and dyes used in this study are listed in the Reagents and Tools.

### Statistical analysis
Statistical analysis was performed using GraphPad Prism software (GraphPad, version 10.2.1). Measures of pooled data, from at least three independent biological replicates, are represented by mean or median and SD or SEM, as indicated in the figure legends. All data sets were subjected to normality tests (Anderson-Darling, D'Agostino and Pearson, Kolmogorov–Smirnov, or Shapiro–Wilk tests) when applicable. Statistical significance was determined using one-way ANOVA with Welch's correction (unequal variances) with Dunnett's T3 post hoc test; RM one-way ANOVA with Geisser-Greenhouse's correction (unequal variances) with Tukey's post hoc test; Ordinary two-way ANOVA Tukey's post hoc test; Multiple t-tests with the Holm–Sidak method's correction with Dunn's post hoc test; and for non-Gaussian datasets using Kruskal–Wallis test with Dunn's post hoc test, as indicated in the figure legends. $p$ values are reported in figure legends and as source data. A $p$ value ≤0.05 was considered statistically significant. The number of independent biological replicates, the number of single cells analyzed, sample conditions and statistical test used for each dataset are indicated in the corresponding figure legends. For all assays, a minimum of three independent biological replicates was initially used per assay to allow statistical evaluation, and the resulting variation was used to determine the adequate number of additional replicates for subsequent experiments. Investigators were not blinded during sample allocation or analyses. However, to minimize selection bias during image acquisition, the operator acquired the first cell encountered in each field of view without pre-selection based on morphology or signal intensity, and automated image analysis was employed where applicable. Inclusion/exclusion criteria were defined prior to data collection. Samples were excluded only in cases of clear technical failure (e.g., insufficient staining specificity or transfer efficiency in immunoblot analysis, as assessed using appropriate positive and negative controls). For larger datasets of single-cell measurements (typically ≈30 cells per condition and ≥3 biological replicates), outliers were excluded where appropriate using the automated ROUT method ($Q = 1\%$) implemented in GraphPad Prism; no data points were manually removed.

## Software

Data collection used the following software: Microscopy: ZEN Blue 2.6 (Zeiss, Germany); Immunoblot: Amersham ImageQuant™ 800 control software (Cytiva, USA); qRT-PCR: CFX Opus 96 (Bio-Rad, USA). For data analysis, the following softwares were used: GraphPad PRISM, version 10.2.1, ImageJ/Fiji, JPK SPM Data Processing Software v8.0.1 (Bruker, Nano, Germany).

## Data availability

RNA-seq data have been deposited in GEO (Gene Expression Omnibus) under accession number GSE297747, available via https://www.ncbi.nlm.nih.gov/geo/query/acc.cgi?acc=GSE297747.

The source data of this paper are collected in the following database record: biostudies:S-SCDT-10_1038-S44319-025-00583-6.

## Peer review information

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

## Acknowledgements

We thank the animal facility of the Faculty of Medicine, Saarland University, for important services, the Robert Ernst lab (Medical Biochemistry and Molecular Biology) for access to the AxioObserver Z1 microscope, the Department of Cellular Neurophysiology (CIPMM, Saarland University) for kindly providing access to the plasma cleaner, and Dr. Nicole Ludwig for access to a Bioanalyzer for RIN measurements. We further thank all members of the Iden laboratory for stimulating discussions. This project was funded by the Deutsche Forschungsgemeinschaft (DFG, German Research Foundation), Projektnummer 200049484; (SFB 1027 projects A12 to SI, B2 to MB, and ZX to VH), Projektnummer 519828155 (DFG INST 256/583-1 FUGG) and Projektnummer 466742891 (DFG INST 256/555-1 FUGG).

## Author contributions

**Carole Luthold**: Conceptualization; Formal analysis; Investigation; Methodology; Validation; Visualization; Writing—first concept; Writing—original draft; Review and editing of manuscript. **Marie Didion**: Formal analysis; Investigation; Methodology; Validation. **Vanessa Samira Rácz**: Formal analysis; Investigation; Methodology; Validation; Review and editing of manuscript. **Emilio Benedum**: Formal analysis; Investigation; Methodology; Validation. **Ann-Kathrin Burkhart**: Methodology; Resources, Review and editing of manuscript. **Nina Demmerle**: Investigation. **Evelyn Wirth**: Investigation. **Gubesh Gunaratnam**: Formal analysis; Investigation; Methodology; Validation. **Sudharshini Thangamurugan**: Formal analysis; Investigation; Methodology; Visualization. **Volkhard Helms**: Validation; Resources; Review and editing of manuscript. **Markus Bischoff**: Resources; Review and editing of manuscript. **Annika Ridzal**: Methodology; Resources. **Sandra Iden**: Conceptualization; Formal analysis; Funding acquisition; Investigation; Methodology; Project administration; Validation; Visualization; Resources; Writing—original draft; Review and editing of manuscript.

Source data underlying figure panels in this paper may have individual authorship assigned. Where available, figure panel/source data authorship is listed in the following database record: biostudies:S-SCDT-10_1038-S44319-025-00583-6.

## Funding

## Disclosure and competing interests statement

The authors declare no competing interests.

# Expanded View Figures

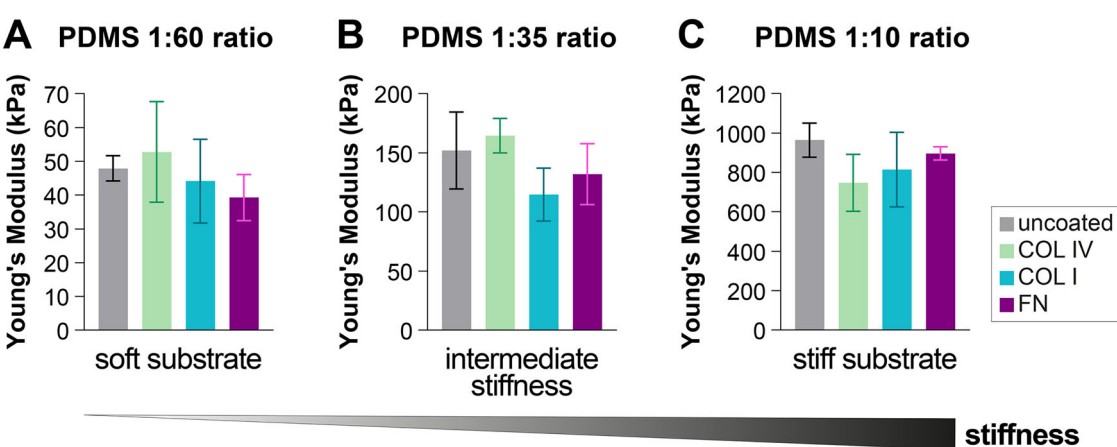

**Figure EV1. Validation of PDMS elasticity using atomic force microscopy.**

Young's modulus of PDMS substrates with varying crosslinker ratios. (A) PDMS substrate with a crosslinker:silicone base agent ratio of 1:60; means ± SD, $N = 3$ (biological replicates); Welch ANOVA tests: ns $p = 0.989$ (uncoated vs. COL I), ns $p = 0.4616$ (uncoated vs. FN), ns $p = 0.9842$ (uncoated vs. COL IV), ns $p = 0.982$ (COL I vs. FN), ns $p = 0.9505$ (COL I vs. COL IV), ns $p = 0.6723$ (FN vs. COL IV). (B) PDMS substrate with a 1:35 ratio; means ± SD, $N \geq 3$ (biological replicates); Welch ANOVA tests: ns $p = 0.2933$ (uncoated vs. COL I), ns $p = 0.8869$ (uncoated vs. FN), ns $p = 0.9686$ (uncoated vs. COL IV), ns $p = 0.9035$ (COL I vs. FN), ns $p = 0.0731$ (COL I vs. COL IV), ns $p = 0.5114$ (FN vs. COL IV). (C) PDMS substrate with a 1:10 ratio; means ± SD, $N = 3$ (biological replicates); Welch ANOVA tests: ns $p = 0.7579$ (uncoated vs. COL I), ns $p = 0.7492$ (uncoated vs. FN), ns $p = 0.3624$ (uncoated vs. COL IV), ns $p = 0.9483$ (COL I vs. FN), ns $p = 0.9939$ (COL I vs. COL IV), ns $p = 0.5694$ (FN vs. COL IV).

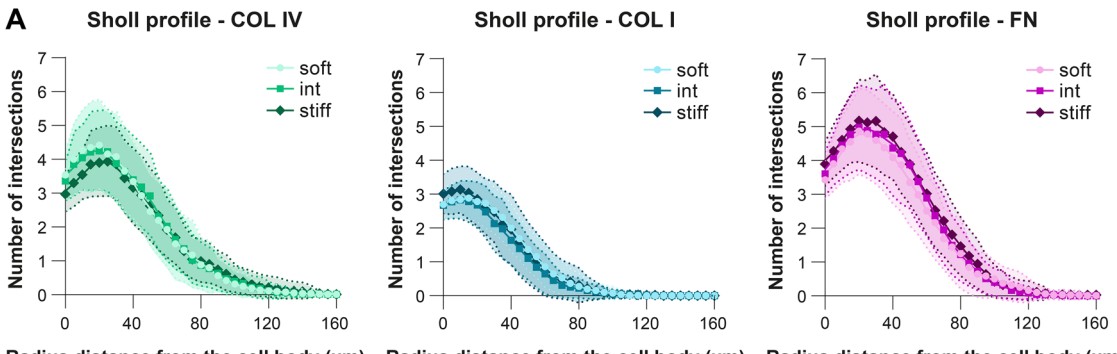

**Figure EV2. Sholl profiles for iMCs exposed to different ECM molecules.**

(A) Graphs showing the Sholl profile analysis, plotting the number of dendrite intersections against the distance from the cell body. The SEM are represented by the connecting curve (dotted line); $N = 3$ (biological replicates), n(cells) ≥85. Each curve represents a substrate stiffness condition, and each graph represents an ECM type (same cells as analyzed in Fig. 1C, different plotting of the data); Welch ANOVA test: COL IV: ns $p > 0.9999$ (1:60 vs. 1:35), ns $p = 0.9650$ (1:60 vs. 1:10), ns $p = 0.9489$ (1:35 vs. 1:10); COL I: ns $p = 0.5182$ (1:60 vs. 1:35), ns $p > 0.9999$ (1:60 vs. 1:10), ns $p = 0.5964$ (1:35 vs. 1:10); FN: ns $p = 0.8906$ (1:60 vs. 1:35), ns $p = 0.4342$ (1:60 vs. 1:10), ns $p = 0.7603$ (1:35 vs. 1:10).

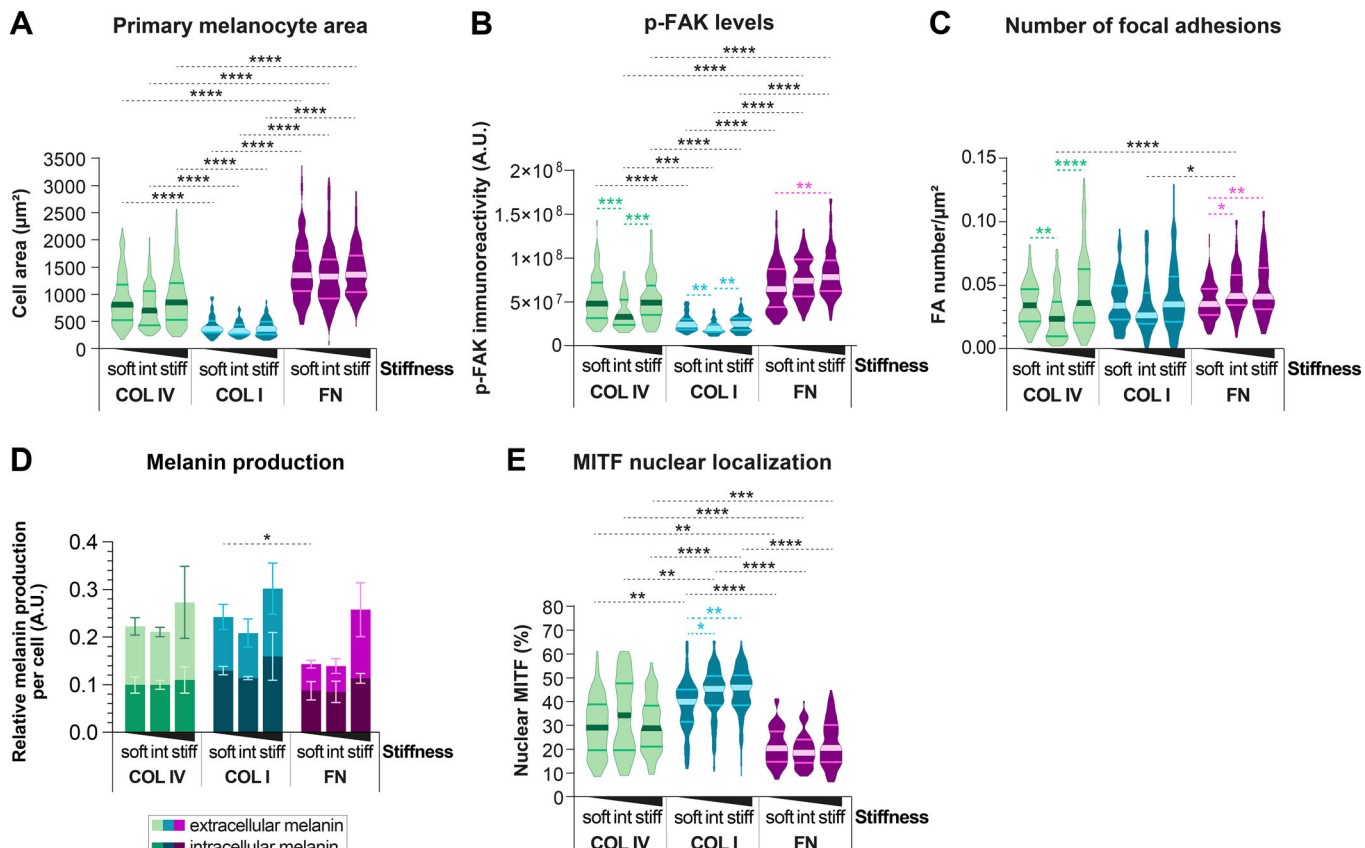

**Figure EV3.  Response of primary MCs to distinct ECM cues.**

pMCs were cultured overnight on substrates of varying stiffness (soft, intermediate or stiff) coated with COL IV, COL I, or FN. (A) Violin plots showing the medians and distributions of the cell area of pMCs; $N \geq 5$ (biological replicates), $n$(cells) $\geq 96$; Kruskal–Wallis test: COL IV: ns $p = 0.5372$ (1:60 vs. 1:35), ns $p > 0.9999$ (1:60 vs. 1:10), $p = 0.1807$ (1:35 vs. 1:10); COL I: ns $p = 0.1552$ (1:60 vs. 1:35), ns $p = 0.7629$ (1:60 vs. 1:10), ns $p > 0.9999$ (1:35 vs. 1:10); FN: ns $p = 0.4268$ (1:60 vs. 1:35), ns $p > 0.9999$ (1:60 vs. 1:10), ns $p = 0.0972$ (1:35 vs. 1:10); 1:60: ****$p < 0.0001$ (COL I vs. FN), ****$p < 0.0001$ (COL I vs. COL IV), ****$p < 0.0001$ (FN vs. COL IV); 1:35: ****$p < 0.0001$ (COL I vs. FN), ****$p < 0.0001$ (COL I vs. COL IV), ****$p < 0.0001$ (FN vs. COL IV); 1:10: ****$p < 0.0001$ (COL I vs. FN), ****$p < 0.0001$ (COL I vs. COL IV), ****$p < 0.0001$ (FN vs. COL IV). (B) Quantification of p-FAK levels; violin plots showing the medians and distributions of the integrated density of total p-FAK per cell; $N = 3$ (biological replicates), $n$(cells) $\geq 52$; Kruskal–Wallis test: COL IV: ***$p = 0.0004$ (1:60 vs. 1:35), ns $p > 0.9999$ (1:60 vs. 1:10), ***$p = 0.0005$ (1:35 vs. 1:10); COL I: **$p = 0.002$ (1:60 vs. 1:35), ns $p > 0.9999$ (1:60 vs. 1:10), **$p = 0.0024$ (1:35 vs. 1:10); FN: ns $p = 0.3084$ (1:60 vs. 1:35), **$p = 0.0035$ (1:60 vs. 1:10), ns $p = 0.7979$ (1:35 vs. 1:10); 1:60: ****$p < 0.0001$ (COL I vs. FN), ****$p < 0.0001$ (COL I vs. COL IV), ns $p = 0.1$ (FN vs. COL IV); 1:35: ****$p < 0.0001$ (COL I vs. FN), ***$p = 0.0003$ (COL I vs. COL IV), ****$p < 0.0001$ (FN vs. COL IV); 1:10: ****$p < 0.0001$ (COL I vs. FN), ****$p < 0.0001$ (COL I vs. COL IV), ****$p < 0.0001$ (FN vs. COL IV). (C) Quantification of FAs; violin plots showing the medians and distributions of the number of focal adhesions per $\mu m^2$ per cell; $N = 3$ (biological replicates), $n$(cells) $\geq 65$; Kruskal–Wallis test: COL IV: **$p = 0.0081$ (1:60 vs. 1:35), ns $p = 0.507$ (1:60 vs. 1:10), ****$p < 0.0001$ (1:35 vs. 1:10); COL I: ns $p = 0.7166$ (1:60 vs. 1:35), ns $p > 0.9999$ (1:60 vs. 1:10), ns $p = 0.1991$ (1:35 vs. 1:10); FN: *$p = 0.0117$ (1:60 vs. 1:35), **$p = 0.0071$ (1:60 vs. 1:10), ns $p > 0.9999$ (1:35 vs. 1:10); 1:60: ns $p > 0.9999$ (COL I vs. FN), ns $p > 0.9999$ (COL I vs. COL IV), ns $p > 0.9999$ (FN vs. COL IV); 1:35: *$p = 0.0167$ (COL I vs. FN), ns $p = 0.5377$ (COL I vs. COL IV), ****$p < 0.0001$ (FN vs. COL IV); 1:10: ns $p > 0.9999$ (COL I vs. FN), ns $p > 0.9999$ (COL I vs. COL IV), ns $p > 0.9999$ (FN vs. COL IV). (D) Quantification of intra- and extracellular melanin content by spectrophotometry at 405 nm from pMCs cultured 72 h on substrates of varying stiffness (soft, intermediate or stiff) coated with COL IV, COL I or FN; means ± SD, $N = 3$ (biological replicates); multiple $t$- tests: Soft vs. Intermediate: ns $p = 0.4947$ (COL I), ns $p = 0.8479$ (FN), ns $p = 0.8479$ (COL IV); Soft vs. Stiff: ns $p = 0.619892$ (COL I), ns $p = 0.10379$ (FN), ns $p = 0.619892$ (COL IV); Intermediate vs. Stiff: ns $p = 0.361807$ (COL I), ns $p = 0.118759$ (FN), ns $p = 0.361807$ (COL IV); COL I vs. COL IV: ns $p = 0.868243$ (1:60), ns $p = 0.935696$ (1:35), ns $p = 0.935696$ (1:10), COL IV vs. FN: ns $p = 0.053003$ (1:60), ns $p = 0.053003$ (1:35), ns $p = 0.836335$ (1:10); COL I vs. FN: *$p = 0.0139$ (1:60), ns $p = 0.0797$ (1:35), ns $p = 0.5564$ (1:10). (E) Quantification of nuclear MITF; violin plots showing the medians and distributions of the percentage of nuclear MITF per cell; $N = 3$ (biological replicates), $n$(cells) $\geq 67$; Kruskal–Wallis test: COL IV: ns $p = 0.2081$ (1:60 vs. 1:35), ns $p > 0.9999$ (1:60 vs. 1:10), ns $p = 0.3965$ (1:35 vs. 1:10); COL I: *$p = 0.0186$ (1:60 vs. 1:35), **$p = 0.0034$ (1:60 vs. 1:10), ns $p > 0.9999$ (1:35 vs. 1:10); FN: ns $p = 0.7181$ (1:60 vs. 1:35), ns $p > 0.9999$ (1:60 vs. 1:10), ns $p = 0.2589$ (1:35 vs. 1:10); 1:60: ****$p < 0.0001$ (COL I vs. FN), **$p = 0.0056$ (COL I vs. COL IV), **$p = 0.0013$ (FN vs. COL IV); 1:35: ****$p < 0.0001$ (COL I vs. FN), **$p = 0.0046$ (COL I vs. COL IV), ****$p < 0.0001$ (FN vs. COL IV); 1:10: ****$p < 0.0001$ (COL I vs. FN), ****$p < 0.0001$ (COL I vs. COL IV), ***$p = 0.0009$ (FN vs. COL IV). AU, arbitrary units.

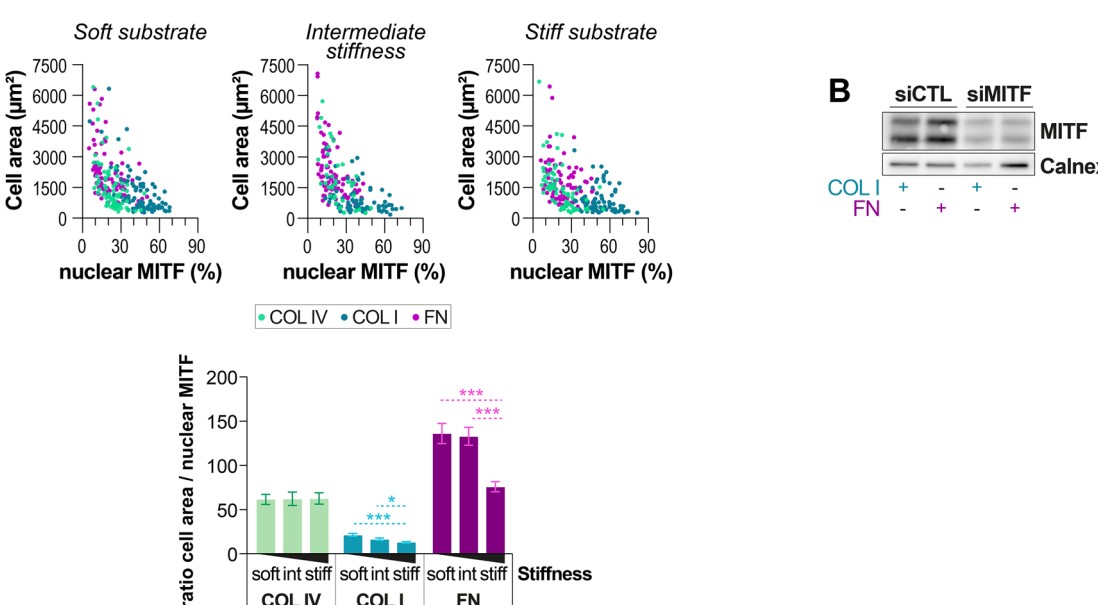

**A**  Correlation between nuclear MITF and cell area

*Soft substrate*

*Intermediate stiffness*

*Stiff substrate*

● COL IV ● COL I ● FN

**Figure EV4. Negative correlation between nuclear MITF levels and MC area and siMITF validation.**

(A) The percentage of nuclear MITF was plotted against the cell area of the corresponding MC for all stiffnesses and ECM types tested. The bar diagram depicts the means ± SEM of the ratio of cell area to percentage of nuclear MITF per cell; $N = 3$ (biological replicates), $n$(cells) ≥47; Kruskal–Wallis test: COL IV: ns $p > 0.9999$ (1:60 vs. 1:35), ns $p = 0.5839$ (1:60 vs. 1:10), ns $p > 0.9999$ (1:35 vs. 1:10); COL I: ns $p = 0.5497$ (1:60 vs. 1:35), ***$p = 0.0001$ (1:60 vs. 1:10), *$p = 0.0329$ (1:35 vs. 1:10); FN: ns $p > 0.9999$ (1:60 vs. 1:35), ***$p = 0.0001$ (1:60 vs. 1:10), ***$p = 0.0001$ (1:35 vs. 1:10); 1:60: ****$p < 0.0001$ (COL I vs. FN), ****$p < 0.0001$ (COL I vs. COL IV), **$p = 0.0014$ (FN vs. COL IV); 1:35: ****$p < 0.0001$ (COL I vs. FN), ****$p < 0.0001$ (COL I vs. COL IV), ***$p = 0.0006$ (FN vs. COL IV); 1:10: ****$p < 0.0001$ (COL I vs. FN), ****$p < 0.0001$ (COL I vs. COL IV), ns $p = 0.0603$ (FN vs. COL IV). (B) Western blot analysis of MITF expression in iMCs plated on COL I or FN and transfected for 72 h with siCtrl or siMITF, performed in parallel with melanin quantification shown in Fig. 4D.

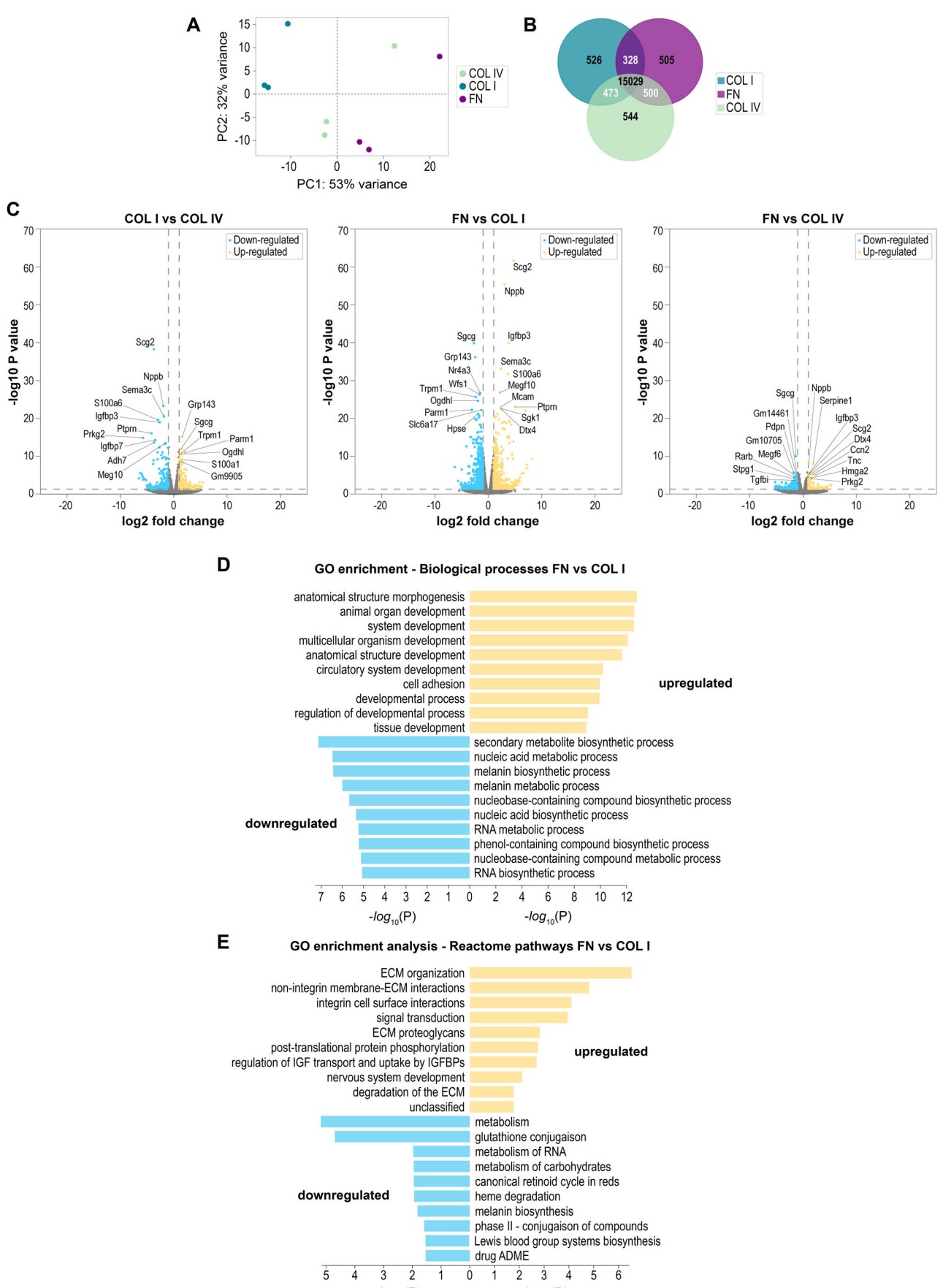

◀ **Figure EV5.  Transcriptomic profiling across ECM conditions.**

iMCs were cultured overnight on stiff substrates coated with COL IV, COL I, or FN; $N = 3$ (biological replicates). (**A**) Principal component analysis (PCA) of gene expression across substrates. Each dot represents one sample: COL IV (green), COL I (blue), and FN (purple). Axes indicate principal components capturing the highest variance. (**B**) Venn diagram showing the number of genes expressed in the samples grown on COL IV, COL I, and FN substrates. (**C**) Volcano plots depicting differentially expressed genes between substrates: (i) COL I vs. COL IV, (ii) FN vs. COL I, and (iii) FN vs. COL IV ($p$ values derived from Wald tests in DESeq2 with Benjamini–Hochberg adjustment). Significantly downregulated genes are shown in blue; upregulated genes in yellow. (**D**) Gene Ontology (GO) Biological Process enrichment analysis of significantly regulated genes across substrates. The x-axis indicates the adjusted $p$ value of Fisher's Exact test; the y-axis lists the most significantly enriched biological processes. (**E**) Reactome pathway enrichment analysis performed on differentially expressed genes across ECM conditions. The x-axis represents the adjusted $p$ value of Fisher's exact test; the y-axis lists the top significantly enriched Reactome pathways.

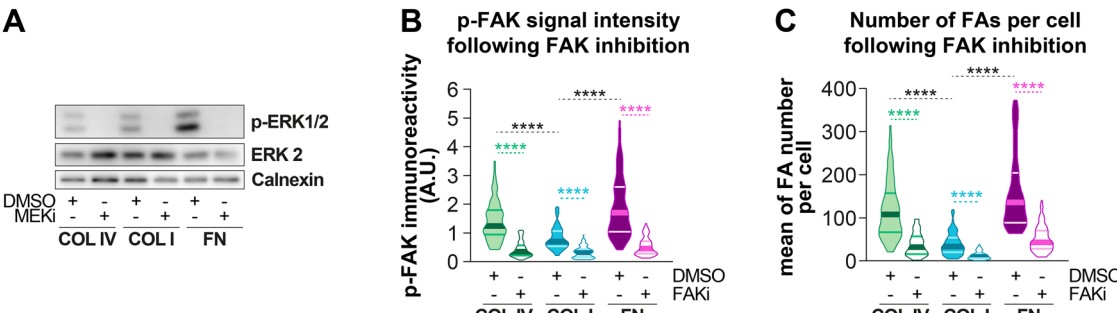

**Figure EV6.   Functional validation of MEK and FAK inhibitors in iMCs cultured on ECM-coated substrates.**

iMCs were cultured on stiff substrates coated with COL IV, COL I, or FN. Four hours post-plating, a 16-h treatment with DMSO, 100 nM of MEK inhibitor (MEKi = Trametinib) or FAK inhibitor (FAKi = Ifebemtinib) was commenced. (A) Representative Western blot showing the inhibition of ERK1/2 phosphorylation with Trametinib treatment. (B) Quantification of p-FAK (Y397) intensity per cell following treatment with DMSO or FAK inhibitor (FAKi); violin plots display medians and distributions; $N = 3$ (biological replicates), $n$(cells) ≥88; Kruskal–Wallis test: DMSO: ****$p < 0,0001$ (COL IV vs. COL I), ns $p > 0,9999$ (COL IV vs. FN), **** $p < 0,0001$ (COL I vs. FN); DMSO vs. FAKi: **** $p < 0,0001$ (COL IV DMSO vs. COL IV FAKi 24 h), ****$p < 0,0001$ (COL I DMSO vs. COL I FAKi 24 h), ****$p < 0,0001$ (FN DMSO vs. FN FAKi 24 h). (C) Quantification of focal adhesion (FA) number per cell following FAKi treatment; violin plots show medians and distributions; $N = 3$ (biological replicates), $n$(cells) ≥70; Kruskal–Wallis test: DMSO: ****$p < 0.0001$ (COL IV vs. COL I), ns $p = 0.6525$ (COL IV vs. FN), ****$p < 0.0001$ (COL I vs. FN); DMSO vs. FAKi: ****$p < 0.0001$ (COL IV DMSO vs. COL IV FAKi), ****$p < 0.0001$ (COL I DMSO vs. COL I FAKi), ****$p < 0.0001$ (FN DMSO vs. FN FAKi).

