## [Peer Review File · EMBO Reports]

Melanocyte differentiation and mechanosensation are differentially modulated by distinct extracellular matrix proteins

Carole Luthold, Marie Didion, Emilio Benedum, Ann-Kathrin Burkhart, Nina Demmerle, Evelyn Wirth, Gubesh Gunaratnam, Vanessa Racz, Sudharshini Thangamurugan, Volkhard Helms, Markus Bischoff, Annika Ridzal, and Sandra Iden

Corresponding author(s): Sandra Iden (sandra.iden@uni-saarland.de), Carole Luthold (carole.luthold@uni-saarland.de)

Review Timeline:

Transfer Date:	4th Jun 25
Editorial Decision:	18th Jul 25
Revision Received:	21st Aug 25
Accepted:	4th Sep 25

Editor: Deniz Senyilmaz Tiebe / Martina Rembold

Transaction Report: This manuscript was transferred to EMBO reports following peer review at Review Commons.

**Review
COMMONS**

Review #1

1. Evidence, reproducibility and clarity:

Evidence, reproducibility and clarity (Required)

****Summary:****

In this manuscript, authors demonstrated the role of ECM-dependent MEK/ERK/MITF signaling pathway that influences the plasticity of MCs (melanocytes) through their interactions with the environment. The findings emphasize the essential role of the extracellular matrix (ECM) in controlling MC function and differentiation, highlighting a critical need for further research to understand the complex interactions between mechanical factors and ECM components in the cellular microenvironment. Overall, the manuscript is concise, written well and shed light on a complex relationship between ECM protein types and substrate stiffness that affects MC mechanosensation. However, understanding detailed molecular mechanisms involved, especially the roles of MITF and other key regulators, is crucial for comprehending MC function and related pathologies. Authors needs to clarify some minor queries to be considered for publication.

****Major comment:****

1. Authors have chosen ERK signaling pathways to test and draw their conclusion based on existing knowledge in the field, as several studies previously reported the role of ECM to modulate the ERK signaling pathway but it would be interesting to test other signaling pathways unbiasedly; e.g. ECM can also regulate Wnt signaling (PMID: 29454361) and connection of MITF and its target gene TYR expression is also regulated by Wnt in context of melanocyte. (PMID: 29454361, PMID: 34878101, PMID: 38020918).
2. Discussion line 340-344. Please provide the data as it is directly connected to the study, and it would be crucial to interpret data better. As FAK is upregulated and FAK inhibitor did not reduce pERK, is there any possibility that other kinases might involve. Please discuss. Again, authors should check Wnt activation as FAK can activate Wnt signaling in response to matrix stiffness as well. (PMID 29454361).
3. Rationale for selecting MITF for the study is very weak. Please justify in the discussion why authors have chosen to study MITF/ERK axis with a more logistic approach.
4. It is suggested to check for the changes in the transcriptomic profile of melanocytes upon culturing on different matrix to get a more comprehensive view associated with the molecular mechanisms involved.
5. Please provide the protein expression of genes involved in cell cycle progression and/or

apoptosis to support the data in Fig. 3D-E.

****Minor comment:****

1. Discussion line 358-359, using term synergy is an overstatement as the collective data do not support the claim. Very little role of matrix stiffness is demonstrated by experimental data.
2. Method section, BrdU assay and BrdU assay-cell proliferation can be combined in method section.
3. What trigger melanocytes to respond to different microenvironment. Please discuss.
4. Fig 3C and 5D Tyr mRNA expression is tested. Authors should also test for the protein expression in the similar set of studies.
5. Line 217-218, Authors claim stiffness mediated increase of MITF nuclear localization in Col I, however Fig. 4A-B does not represent that claim. Please justify.

2. Significance:

Significance (Required)

Overall, the study is well-planned, the experiments are well-designed and executed with appropriate use of statistical analysis. However, a more in-depth analysis of the molecular mechanisms is necessary to clarify how the extracellular matrix (ECM) regulates ERK or MITF nuclear translocation.

This study enhances our existing knowledge by linking the well-established role of the extracellular matrix (ECM) in regulating ERK signaling to ERK's involvement in controlling MITF, a key regulator of melanocyte differentiation. It further establishes the ECM's role in controlling melanocyte function and differentiation.

This study will interest readers working in the field of the tumor microenvironment, as it explores the role of the extracellular matrix and its complexity and stiffness in disease progression, not only in melanoma but also in other types of cancer.

3. How much time do you estimate the authors will need to complete the suggested revisions:

Estimated time to Complete Revisions (Required)

(Decision Recommendation)

Between 1 and 3 months

4. Review Commons values the work of reviewers and encourages them to get credit for their work. Select 'Yes' below to register your reviewing activity at Web of Science Reviewer Recognition Service (formerly Publons); note that the content of your review will not be visible on Web of Science.

Yes

Review #2

1. Evidence, reproducibility and clarity:

Evidence, reproducibility and clarity (Required)

****Summary****

In their manuscript, Luthold et al describe the behaviour of immortalized mouse melanocytes cultured on various extracellular matrix (ECM) proteins and substrates of different stiffness. They found that fibronectin, collagen IV and collagen I have different effects on melanocyte morphology, migration, and proliferation. They further link these differential effects to MITF localization and MEK/ERK signalling. This work shows that fibronectin supports melanocyte migration, which was associated with a dendritic morphology and correlated with increased MEK/ERK signalling and decreased MITF nuclear localization. In contrast, collagen I promoted melanocyte proliferation with low MEK/ERK signalling, enhanced MITF nuclear localization and high melanin production.

While this study is well designed and the data adequately presented and interpreted, the impact of its conclusions is limited by the incomplete mechanistic characterization of the observed phenotypes and by the lack of parallels with physiological conditions. To strengthen their manuscript, the authors should consider the following comments:

****Major comments****

1. Characterization of observed phenotypes:

The link between matrix-sensing and intracellular signalling is missing. Which types of integrins are expressed by iMCs? Are any of these integrins required for the observed phenotypes?

The phenotypic changes described here are interesting but only partially analysed. Transcriptomic studies would yield a more complete view of cell state transitions

(optional). At a minimum, could the authors detect any changes in cadherin expression, or in other genes classically involved in phenotype switching, such as twist1, snail or zeb1? Lines 235-236, the authors write that ECM proteins regulate melanocyte behaviour "likely through modulation of MITF localization and activity". Could the authors support the role of MITF experimentally? Genetic experiments using different MITF mutants could address this question.

Additionally, how does MEK/ERK signalling control MITF activity in these melanocytes? The trametinib experiment should be consolidated with other inhibitors (including ERK inhibitors) and/or genetic manipulation. Did the authors also measure the effect of trametinib on cell proliferation in Figure 5?

2. Parallels with physiological conditions:

Most experiments shown were performed with immortalized melanocytes even though authors mention the use of primary cells (pMCs, line 148). Were similar results obtained in primary melanocytes?

Do human melanocytes in culture behave similarly? Are some of these observations also true in vivo, for example in mouse skin (optional)?

How do the authors reconcile their findings that collagen IV induced melanocyte migration and decreased proliferation and melanin production with the fact that melanocytes in human skin are generally in contact with the collagen IV-rich basement membrane?

****Minor comment****

The evidence that FAK is not responsible for MEK/ERK activation could be presented in the main text rather than in the discussion.

2. Significance:

Significance (Required)

General assessment: This study establishes the cellular impact of different types of extracellular matrix proteins and stiffness conditions relevant to skin biology on the behaviour of untransformed mouse melanocytes. In particular, it shows opposite effects of fibronectin and collagen I on cell proliferation and migration, which could prove relevant to certain skin conditions in human. However, the scope of these results is limited by the incomplete mechanistic characterization of the observed phenotypes and by the lack of parallels with physiological conditions.

Advance: The systematic comparison of different microenvironmental conditions on

normal melanocyte behaviour is novel and opens perspectives to understand the role of melanocytes in some human skin pathologies.

Audience: The comparison of different environmental conditions on melanocyte behaviour is of interest to the melanocyte biology community and could have implications for basic and clinical understanding of some skin diseases.

My expertise is in melanoma biology, including the impact of the microenvironment on tumour cell behaviours.

3. How much time do you estimate the authors will need to complete the suggested revisions:

Estimated time to Complete Revisions (Required)

(Decision Recommendation)

Between 1 and 3 months

4. Review Commons values the work of reviewers and encourages them to get credit for their work. Select 'Yes' below to register your reviewing activity at Web of Science Reviewer Recognition Service (formerly Publons); note that the content of your review will not be visible on Web of Science.

Yes

Review #3

1. Evidence, reproducibility and clarity:

Evidence, reproducibility and clarity (Required)

In this manuscript Luthold et al. describe how extracellular matrix proteins and mechanosensation affect melanocyte differentiation. In particular, they show that ECM proteins and surface stiffness lead to effects on the MEK/ERK pathway, thus affecting the MITF transcription factor. The manuscript is interesting, well written and the data presented in a clear and easy-to-follow manner. The data are nicely quantitated and largely convincing.

However, the discussion of the nuclear location of MITF (Figure 4A) is not convincing. The images presented show that upon exposure to Coll, there is a lot of MITF in the nucleus, a lot less so upon CollIV and none upon FN exposure. However, we only see a snapshot of the

cells and thus we do not know if we are witnessing effects on MITF protein synthesis, degradation or nuclear localization (the least likely scenario since M-MITF, the isoform present in melanocytes is predominantly nuclear anyway). Was there a cytoplasmic signal detected? Upon FN treatment, there is no MITF protein visible in the cells. Does this mean that the protein is not made, that it is degraded or present at such low levels that the antibody does not detect it? The claim of the authors that this affects nuclear localization of MITF needs more corroboration. Also, the authors need to show immunocytochemical images for the effects on MITF nuclear localization for the images presented in Figure 5C. It seems that the authors quantitated immune-reactivity for both MITF and YAP. What was the control and how was the data normalized? Similarly, the blots and data shown in Figure 5 are not consistent with the text as described in the results section. The differences observed are minor and the only set that is likely to be significant is the FN-set; the differences between soft, intermediate and stiff of the FN-set do not look significantly different. The description of this in the results section should be toned down accordingly.

2. Significance:

Significance (Required)

Upon improvement, this paper will provide an early characterization of the effects of the ECM on melanocyte differentiation. If the link to MITF holds, this will be the first time that mechanosensation has been shown to mediate effects on this transcription factor.

3. How much time do you estimate the authors will need to complete the suggested revisions:

Estimated time to Complete Revisions (Required)

(Decision Recommendation)

Between 1 and 3 months

Yes

Full Revision

Manuscript number: RC-2024-02729

Corresponding author(s): Carole Luthold and Sandra Iden

1. General Statements [optional]

We thank the reviewers for the time and caution with which they evaluated our work. We are grateful for their constructive feedback, which has helped to improve the quality of the manuscript, and are pleased that the importance and quality of our work has been recognized. We performed new experiments to further substantiate our conclusions and comprehensively address all reviewers' concerns, as outlined in our detailed point-by-point responses.

First, however, we would like to place our work in a general context and mention the major findings of the revised manuscript. Deciphering mechanisms that govern the differentiation and function of melanocytes, the pigment-producing cells responsible for skin photoprotection, is critical to understand processes that mediate pigmentation disorders and malignant melanoma. Melanocyte differentiation is orchestrated by the melanocyte-inducing transcription factor (MITF), which controls the expression of pigmentation genes and maintains melanocyte homeostasis. Previous reports in different cell systems revealed important roles of extracellular matrix (ECM) cues, such as matrix protein composition and substrate stiffness, in influencing cellular behaviors such as proliferation and differentiation. However, at present it is unclear how these extrinsic factors affect MITF regulation and melanocyte functions.

In this revised manuscript, we present novel insights into the **effect of distinct ECM components and substrate stiffness in regulating melanocyte differentiation and function through a FAK-ERK-MITF signaling axis**. We exposed primary murine melanocytes and a melanocyte cell line generated in our lab to substrates of varying stiffness and coated with different ECM proteins. Subsequent quantitative imaging, transcriptome and signal transduction analyses as well as pharmacological inhibition of pathways that responded to these extrinsic cues revealed four major new findings:

1. **We identified MITF as a novel ECM- and mechanosensitive transcription factor whose nuclear localization is fine-tuned by ECM components and stiffness.** We showed that MITF nuclear localization is enhanced on type I collagen (COL I) compared to type IV collagen (COL IV), with substrate stiffness further increasing this effect on COL I. In contrast, fibronectin (FN) exposure led to significantly reduced nuclear MITF levels compared to both COL I and COL IV. Our revision experiments delineated that ECM composition influences MITF primarily by altering its localization, and siRNA-mediated depletion confirmed the essential role of MITF as an effector of ECM-derived signals in MCs.
2. **ECM-dependent MITF localization is controlled by MEK/ERK activity and elicits a shift in melanocyte differentiation.** We found that ECM components differentially regulate ERK activity, which in turn controls MITF nuclear localization and initiates distinct melanocyte differentiation phenotypes: Compared to FN, COL I restricted ERK activity, resulting in elevated levels of nuclear MITF, increased melanin production and proliferation but restricted migration.

In contrast, FN stimulated ERK activity, resulting in low nuclear MITF and a dedifferentiated, slow-cycling and motile phenotype with reduced pigmentation. Pharmacological MEK or ERK inhibition effectively reverted the low nuclear MITF and melanin production observed in FN-exposed melanocytes. The original data already demonstrated ECM-dependent activation of FAK, and the revised manuscript now places FAK upstream of MEK/ERK in this context. FAK inhibition in FN-exposed MCs reduced ERK activation and enhanced melanin production, supporting a functional FAK–ERK–MITF axis in MCs.

- 3. ECM types elicit significant transcriptomic changes in genes associated with MC pigmentation and differentiation.** To learn how ECM molecules affect MC behavior at the transcriptional level, during revision we conducted bulk RNA sequencing of MCs cultured on COL IV, COL I, or FN. This unbiased approach revealed phenotypic shifts in response to distinct ECM, supporting that ECM molecules modulate gene expression programs related to MC differentiation and pigmentation, and providing a broader molecular context for our findings.
- 4. Combined role of ECM components and stiffness in melanocyte mechanosensation.** We found that the response of melanocytes to mechanical cues is ECM-dependent, and that distinct ECM protein types modulate stiffness responses of MCs. Notably, COL I supports stiffness-induced MITF nuclear localization and proliferation, while FN enhances mechanosensitive responses at the level of focal adhesions and focal adhesion kinase activity. This interaction highlights the importance of considering both ECM composition and mechanical properties when studying cell behavior.

Collectively, our original and new data establish a mechanistic link between ECM cues and MC plasticity through modulation of the FAK–ERK–MITF signaling axis. We show that ECM components impact on ERK activity, which in turn regulates MITF localization, thereby influencing MC differentiation and key functions such as pigmentation and proliferation. By integrating pharmacological, biophysical, cell biological and transcriptomic approaches, our study identifies MITF as a central integrator of ECM-derived signals in MCs.

This work provides a novel framework for understanding how microenvironmental features shape MC differentiation state and behavior, with potential implications for skin physiology and disease.

(answers by authors in blue)

A. Reviewer #1 Evidence, reproducibility and clarity:

Summary:

In this manuscript, authors demonstrated the role of ECM-dependent MEK/ERK/MITF signaling pathway that influences the plasticity of MCs (melanocytes) through their interactions with the environment. The findings emphasize the essential role of the extracellular matrix (ECM) in controlling MC function and differentiation, highlighting a critical need for further research to understand the complex interactions between mechanical factors and ECM components in the cellular microenvironment. Overall, the manuscript is concise, written well and shed light on a complex relationship between ECM protein types and substrate stiffness that affects MC mechanosensation. However, understanding detailed molecular mechanisms involved, especially the roles of MITF and other key regulators, is crucial for comprehending MC function and related pathologies. Authors need to clarify some minor queries to be considered for publication.

We thank this reviewer for the time and caution taken to assess our work. To provide a better understanding of the molecular mechanisms involved in MITF modulation and MC function in response to ECM proteins, we substantially revised the manuscript and now included e.g. bulk RNA sequencing, pharmacological inhibition of FAK and ERK (in addition to MEK inhibition), and MITF depletion.

Major comments to the Authors:

1. Authors have chosen ERK signaling pathways to test and draw their conclusion based on existing knowledge in the field, as several studies previously reported the role of ECM to modulate the ERK signaling pathway but it would be interesting to test other signaling pathways unbiasedly; e.g. ECM can also regulate Wnt signaling (PMID: 29454361) and connection of MITF and its target gene TYR expression is also regulated by Wnt in context of melanocyte. (PMID: 29454361, PMID: 34878101, PMID: 38020918).

The new transcriptome analysis (line 258 ff., revised fig. 5, new fig. 6, new suppl. fig. S5) indeed showed that some components of the Wnt signaling pathway are differentially expressed in response to ECM proteins (new fig. 6B). In comparison, however, the expression of genes involved in MAPK/ERK signaling was more prominently affected by the specific ECM types (new fig. 6C, D), congruent with the biochemical results we presented in the original manuscript. We therefore focused our mechanistic analyses on this pathway, and we consolidated our initial findings with additional pharmacological inhibition experiments. Specifically, like MEK inhibition, ERK inhibition (new fig. 6J-L)

increased both MITF nuclear localization and melanin production in MCs exposed to FN, reinforcing the relevance of this pathway in control of MC functions in the model used.

We agree that an additional contribution of Wnt signaling to ECM-dependent regulation of MC phenotypes is possible, including *Mitf* and *Tyr* expression. Next to the new Wnt-related transcriptome data (**line 323 ff., new fig. 6B**), we therefore now included a short discussion on that aspect (**line 478 ff.**). However, we feel that a comprehensive comparison of the individual contributions of Wnt vs. ERK signaling is beyond the scope of the current manuscript.

2. Discussion line 340-344. *Please provide the data as it is directly connected to the study, and it would be crucial to interpret data better. As FAK is upregulated and FAK inhibitor did not reduce pERK, is there any possibility that other kinases might involve. Please discuss. Again, authors should check Wnt activation as FAK can activate Wnt signaling in response to matrix stiffness as well. (PMID 29454361).*

We agree with the reviewer that the FAK data required further investigation. In the revised version, we re-examined the potential role of FAK as an upstream regulator of ERK activation using the FAK inhibitor Ifebemtinib, rather than Defactinib as used in our original experiments. Our previous conclusion—that ERK activation was independent of FAK—was likely influenced by limitations associated with Defactinib, which did not properly reduce p-FAK levels despite lowering focal adhesion numbers, accompanied with an increase of ERK phosphorylation alongside a decrease of nuclear MITF levels. In contrast, Ifebemtinib treatment led to a more effective inhibition of FAK, as evidenced by a marked reduction in both p-FAK levels and focal adhesion number (**new suppl. fig. 6B,C**). Importantly, this was accompanied by a significant decrease in p-ERK levels (**new fig. 6M,N**), suggesting that FAK contributes to ERK activation in response to ECM molecules in our model. Furthermore, FAK inhibition similar to MEK and ERK inhibition, led to increased melanin production in MCs cultured on FN (**new fig. 6O**). These new data are now included in the revised version of the manuscript (**line 360 ff., new fig. 6M-O, new suppl. fig. 6**).

Nonetheless, this does not exclude the possibility that additional kinases and pathways, including Wnt signaling, may also be involved. We acknowledge this possibility in the **revised discussion (lines 478-488)**.

3. Rationale for selecting MITF for the study is very weak. Please justify in the discussion why authors have chosen to study MITF/ERK axis with a more logistic approach.

We have focused central aspects of our analyses on MITF because it is a central regulator of MC differentiation, pigmentation, and survival, and its activity has previously been reported to be modulated by ERK. Considering the observed changes in pigmentation, proliferation, and gene expression in response to distinct ECM molecules, we hypothesized that MITF acts as a key integrator of these ECM-dependent signals. Our

data indeed support this rationale: we detected ECM-type-dependent MITF levels and localization, and manipulating the ERK pathway altered MITF activity and associated functional outputs. Moreover, siRNA-mediated downregulation of MITF in MCs cultured on COL I led to a marked reduction in melanin content (**revised fig. 4D**). Together, these data emphasize that the ERK/MITF axis serves as a pathway that responds to extracellular cues and links these to MC behavior. For clarity, we have included an additional explanation on our rationale in the revised manuscript (**lines 146-152**).

4. It is suggested to check for the changes in the transcriptomic profile of melanocytes upon culturing on different matrix to get a more comprehensive view associated with the molecular mechanisms involved.

We fully agree with the reviewer on the importance of assessing the ECM-dependent transcriptomes of MCs. Therefore, we have now performed RNA sequencing to compare the transcriptomic profiles of MCs cultured on COL IV-, COL I- and FN-coated stiff substrates (**line 258 ff. and revised fig. 5, new fig. 6, new suppl. fig. S5**). This analysis provided a broader view of the molecular responses of MCs to ECM molecules and complemented our previous molecular and phenotypes analyses. The obtained transcriptomes confirmed significant modulation of genes associated with MC differentiation and pigmentation, as well as genes involved in signaling pathways such as MAPK/ERK and Wnt (see also answers to points 1-3). These findings help contextualize the ECM-dependent phenotypic changes and strengthen the mechanistic insights presented in the study.

5. Please provide the protein expression of genes involved in cell cycle progression and/or apoptosis to support the data in Fig. 3D-E.

To support the observations presented in **original fig. 3**, we employed immunostaining to assess the protein expression of Ki67, which is both a well-established marker and a protein involved in cell cycle progression (PMID: 28630280). In **revised figure 3E**, a significant reduction in the proportion of Ki67-positive cells on FN compared to COL I was observed, reinforcing our initial findings derived from BrdU incorporation assays and direct microscopic monitoring of cell division (**revised fig. 3D,F**).

In addition, global gene expression analysis revealed differentially expressed genes related to cell cycle regulation and apoptosis (**revised fig. 5C,D**), in line with the reduced proliferation observed. Notably, FN also triggered the differential expression of genes associated with cellular senescence (**revised fig. 5E**). Together, these data suggest that adhesion to FN induces a transcriptional and phenotypic shift in MCs toward a less-proliferative state that is associated with differential cell cycle modulation and signs of senescence.

Minor comment to the Authors:

1. Discussion line 358-359, using term synergy is an overstatement as the collective data do not support the claim. Very little role of matrix stiffness is demonstrated by experimental data.

We thank the reviewer for this comment and agree that the term “synergy” may overstate the conclusions drawn from the current dataset. We have therefore removed this term from the revised version of the manuscript to more accurately reflect the data.

2. Method section, BrdU assay and BrdU assay-cell proliferation can be combined in method section.

We have combined the descriptions of the BrdU assay and BrdU-based cell proliferation assay into a single, unified section in the Methods.

3. What trigger melanocytes to respond to different microenvironment. Please discuss.

To address this question, we have added the following paragraph to the **Discussion (lines 377-380)**: “Our study identifies ECM components as critical environmental triggers that instruct MC behavior. Through dynamic interactions with the ECM, MCs engage adhesion-dependent signaling pathways, such as FAK activation, enabling them to decode contextual ECM inputs and adapt their phenotype accordingly.”

4. Fig 3C and 5D Tyr mRNA expression is tested. Authors should also test for the protein expression in the similar set of studies.

We thank the reviewer for this suggestion and agree that assessing TYR protein expression would be valuable. However, we have encountered difficulties with the currently available antibodies and detection methods, which in our hands appeared unreliable for consistently detecting endogenous TYR protein levels in MCs under these conditions. For this reason, we relied on *Tyr* mRNA expression as a robust and reproducible readout and complemented this with functional assays such as melanin content measurement as a read-out that indirectly reflects TYR enzymatic activity. Of note, our transcriptomic analysis also revealed *Tyr* and other melanogenesis genes as differentially expressed genes when comparing MCs grown on COLI vs FN (**revised fig. 5A,B**).

5. Line 217-218, Authors claim stiffness mediated increase of MITF nuclear localization in Col I, however Fig. 4A-B does not represent that claim. Please justify.

Fig. 4A shows representative images of MCs cultured on stiff substrates coated with different ECM types, while the **original figure 4B** included the comparison across substrate stiffnesses for each ECM condition. We have now generated additional datasets

to assess global MITF levels as well as nuclear localization across stiffness conditions in the presence of the different ECM types, demonstrating that nuclear MITF is significantly higher in cells cultured on stiff vs. soft or intermediate stiffness (**revised fig. 4B,C**). Of note, we do not detect a significant difference between soft and intermediate substrate stiffness, which could hint to a threshold of MITF dynamics in stiffness sensitivity. We have updated the figure legend and corresponding text to ensure the data presentation accurately supports our conclusions.

Significance:

Overall, the study is well-planned, the experiments are well-designed and executed with appropriate use of statistical analysis. However, a more in-depth analysis of the molecular mechanisms is necessary to clarify how the extracellular matrix (ECM) regulates ERK or MITF nuclear translocation.

We agree and feel that the additional data in the revised manuscript that explored transcriptional changes and the FAK/MEK/ERK/MITF axis in response to ECM proteins provide improved insights into ECM-mediated regulation of ERK and MC pigmentation.

This study enhances our existing knowledge by linking the well-established role of the extracellular matrix (ECM) in regulating ERK signaling to ERK's involvement in controlling MITF, a key regulator of melanocyte differentiation. It further establishes the ECM's role in controlling melanocyte function and differentiation.

This study will interest readers working in the field of the tumor microenvironment, as it explores the role of the extracellular matrix and its complexity and stiffness in disease progression, not only in melanoma but also in other types of cancer.

B. Reviewer #2 Evidence, reproducibility and clarity:

Summary:

In their manuscript, Luthold et al describe the behaviour of immortalized mouse melanocytes cultured on various extracellular matrix (ECM) proteins and substrates of different stiffness. They found that fibronectin, collagen IV and collagen I have different effects on melanocyte morphology, migration, and proliferation. They further link these differential effects to MITF localization and MEK/ERK signalling. This work shows that fibronectin supports melanocyte migration, which was associated with a dendritic morphology and correlated with increased MEK/ERK signalling and decreased MITF nuclear localization. In contrast, collagen I promoted melanocyte proliferation with low MEK/ERK signalling, enhanced MITF nuclear localization and high melanin production.

While this study is well designed and the data adequately presented and interpreted, the impact of its conclusions is limited by the incomplete mechanistic characterization of the observed phenotypes and by the lack of parallels with physiological conditions. To strengthen their manuscript, the authors should consider the following comments:

We also wish to thank this reviewer for the efforts made to assess our work and help us improve the study. We substantially revised the manuscript and now included e.g. bulk RNA sequencing and various loss-of-function approaches to better delineate the signaling pathways involved in ECM-dependent control of MCs.

Major comments to the Authors:

1. Characterization of observed phenotypes:

- *The link between matrix-sensing and intracellular signalling is missing. Which types of integrins are expressed by iMCs?*

This is indeed an interesting point. Our RNA sequencing analysis indicates that MCs express integrins known to mediate adhesion to COL I and FN, including Itga2, Itga3, Itga5, Itgav, Itgb1, and Itgb3 (**revised fig. 5K**). Importantly, the expression of these integrins remains relatively consistent across ECM conditions (COL I, COL IV, and FN), suggesting that the phenotypic differences observed may not be directly explained by variations in integrin expression.

- *Are any of these integrins required for the observed phenotypes?*

To assess a functional involvement, we conducted a pilot experiment blocking β 1-integrin in MCs seeded on COL I and observed a marked reduction in MC adhesion (see associated graph 1, provided to this reviewer). However, the compromised cell spreading and resulting widespread detachment introduced confounding effects, making it difficult to interpret downstream events such as MITF nuclear localization. Since such readouts can be indirectly influenced by the overall adhesion state and associated signaling pathways such as FAK, we chose not to pursue further mechanistic analysis using this approach. Targeted strategies (e.g., inducible knockdown, acute protein degradation) will be needed in the future to dissect the precise role of individual integrins in mediating ECM-specific signaling responses in MCs.

Graph for referee with unpublished data and its description has been removed upon request by the authors.

- *The phenotypic changes described here are interesting but only partially analysed. Transcriptomic studies would yield a more complete view of cell state transitions (optional). At a minimum, could the authors detect any changes in cadherin expression, or in other genes classically involved in phenotype switching, such as twist1, snail or zeb1?*

We thank the reviewer for this important suggestion, which helped to improve this manuscript. We have now performed bulk RNA sequencing to analyze global gene expression changes in MCs cultured on different ECM substrates (**revised fig. 5, new suppl. fig. 5**). Among these, we explored gene expression programs associated with MC plasticity and differentiation (**revised fig. 5F–H**): MCs cultured on FN exhibited reduced expression of melanocytic differentiation markers and upregulation of genes linked to plasticity, dedifferentiation, and neural crest-like features, suggesting a shift toward a less differentiated state, reflecting aspects of a phenotypic switch.

Nonetheless, as part of this analysis (but not included in the manuscript), we found that *Zeb2*, *Snai2*, and *Zeb1* were expressed at similar levels across ECM conditions. Similarly, among the cadherins, *Cdh1* and *Cdh2* were not differentially expressed, albeit the overall low expression of *Cdh1* showed a trend towards a reduction on COL I. Finally, *Snai1*, *Twist1*, and *Twist2* were detected at very low levels and not significantly regulated as well. These data suggest that, at the chosen experimental conditions, while a clear adaptive phenotypic cell plasticity is observed, classical EMT-like programs are not prominently

activated. However, we cannot exclude the possibility that longer culture durations or additional cues could induce such transitions.

- *Lines 235-236, the authors write that ECM proteins regulate melanocyte behaviour "likely through modulation of MITF localization and activity". Could the authors support the role of MITF experimentally? Genetic experiments using different MITF mutants could address this question.*

To experimentally support the role of MITF, we now performed melanin assays following siRNA-mediated knockdown of MITF in MCs grown on COL I or FN. On COL I-coated substrates, MITF depletion led to a marked reduction in melanin content, supporting the conclusion that ECM-dependent regulation of pigmentation in our culture model involves MITF activity. These findings are now included in the revised manuscript (**lines 244-245, revised fig. 4D, new fig. S4B**).

- *Additionally, how does MEK/ERK signalling control MITF activity in these melanocytes? The trametinib experiment should be consolidated with other inhibitors (including ERK inhibitors) and/or genetic manipulation.*

To address this comment, we complemented our former Trametinib experiments with ERK inhibition using Ravoxertinib (**new fig. 6J-L**). ERK inhibition led to increased nuclear localization of MITF and elevated melanin production, supporting the involvement of MEK/ERK in restraining MITF activity in MCs in response to ECM molecules. These new data are now included in the revised manuscript (**line 354 ff. and new fig. 6J-L**).

- *Did the authors also measure the effect of trametinib on cell proliferation in Figure 5?*

Overall, compared to the observed pronounced phenotypes like ECM-dependent cell morphology, melanin production and others, the differences in cell proliferation of MCs grown at different ECM conditions were statistically significant but not very large. We therefore refrained from additionally assessing the effect of trametinib on the observed ECM-dependent MC behaviour. Given the well-established role of ERK signaling in promoting cell proliferation, we indeed expect that MEK inhibition can reduce MC proliferation in our system, though it remains open whether there is an ECM-specific aspect to this.

2. Parallels with physiological conditions:

- *Most experiments shown were performed with immortalized melanocytes even though authors mention the use of primary cells (pMCs, line 148). Were similar results obtained in primary melanocytes? Do human melanocytes in culture behave similarly?*

While we have not assessed human MCs, **original fig. S2 (revised fig. S3)** provides data using primary murine MCs (freshly isolated from newborn mice), confirming a similar

behavior of primary cells compared to immortalized MCs in terms of cell area, p-FAK levels, number of FAs, melanin production, and MITF nuclear localization.

- *Are some of these observations also true in vivo, for example in mouse skin (optional)?*

The current manuscript focuses on the behavior of MCs in culture, as it was important to use a reductionist model system that can uncouple the effect of distinct ECM types as well as substrate stiffness. However, as a perspective and beyond the scope of this manuscript, we indeed plan to translate our in vitro findings to mouse skin, taking different biophysical and biochemical cues into account. Data from the present in vitro study provides valuable insights into which parameters and which anatomical areas to study in vivo.

- *How do the authors reconcile their findings that collagen IV induced melanocyte migration and decreased proliferation and melanin production with the fact that melanocytes in human skin are generally in contact with the collagen IV-rich basement membrane?*

We indeed regarded the use of collagen IV (COL IV) as a physiological reference condition, and considered MC migration, proliferation, and melanin production on COL as baseline levels. Relative to COL IV, COL I reduced migration and increased melanin production, while FN led to increased migration, and a decrease of proliferation and melanin production. This suggests that ECM composition can selectively modulate distinct aspects of MC behavior compared to attachment to COL IV. The intermediate state observed on COL IV would be in line with a model in which this abundant basement membrane molecule enables MCs to maintain high flexibility in their phenotype, e.g. to further increase melanin production upon external stimuli other than ECM (UV, inflammation etc.). The perhaps unexpected, opposing response of MCs to FN and COL I, respectively, opens the possibility that under specific (patho)physiological conditions, the then abundant ECM can direct MC behaviour. Both plasma- and cellular-derived FN is deposited upon skin injury and instructs various cell types to promote skin repair. Taking our observations in vitro into account, it is tempting to speculate that this FN-enriched tissue enables MCs to quickly migrate into wound sites to re-establish protection to UV. Conversely, increased COL I levels—as observed in fibrotic conditions such as scleroderma—might favor a more differentiated, pigment-producing phenotype. Interestingly, cases of localized hyperpigmentation have been reported in scleroderma patients, possibly reflecting such matrix-driven MC reprogramming. Though requiring further investigation, these observations open new avenues to explore how dynamic changes in ECM composition contribute to MC behavior in tissue homeostasis and repair. We now extended our original discussion to better emphasize the physiological relevance of our findings (**lines 383-391**) and hypothesize how ECM remodeling may contribute to the dynamic regulation of MC plasticity—not only during tissue homeostasis, but also in response to injury and in fibrotic conditions such as scleroderma (**lines 393-406**).

Minor comments to the Authors:

The evidence that FAK is not responsible for MEK/ERK activation could be presented in the main text rather than in the discussion.

We thank the reviewer for highlighting this important point. Our initial conclusion—that ERK activation was independent of FAK—likely stemmed from limitations of the previously used FAK inhibitor (Defactinib). In those earlier experiments, while FAK inhibition reduced focal adhesion numbers, p-FAK levels were not properly decreased, and paradoxically, ERK phosphorylation increased alongside decreased nuclear MITF levels. Based on this initial discrepancy and because of this reviewer's comment, we performed additional experiments using another selective FAK inhibitor, Ifebemtinib, which achieved an effective reduction in both p-FAK levels and focal adhesion number (**new suppl. fig. S6B, C**). In the revised version, we present new experiments using Ifebemtinib, demonstrating that FAK inhibition in fact does reduce p-ERK levels (**new fig. 6M–N**), thus supporting the notion that FAK contributes to ECM-dependent ERK pathway activation in our model. These findings are now shown in the results section (**lines 357-364**).

Significance:

General assessment: This study establishes the cellular impact of different types of extracellular matrix proteins and stiffness conditions relevant to skin biology on the behaviour of untransformed mouse melanocytes. In particular, it shows opposite effects of fibronectin and collagen I on cell proliferation and migration, which could prove relevant to certain skin conditions in human. However, the scope of these results is limited by the incomplete mechanistic characterization of the observed phenotypes and by the lack of parallels with physiological conditions.

Advance: The systematic comparison of different microenvironmental conditions on normal melanocyte behaviour is novel and opens perspectives to understand the role of melanocytes in some human skin pathologies.

Audience: The comparison of different environmental conditions on melanocyte behaviour is of interest to the melanocyte biology community and could have implications for basic and clinical understanding of some skin diseases.

My expertise is in melanoma biology, including the impact of the microenvironment on tumour cell behaviours.

C. Reviewer #3 Evidence, reproducibility and clarity:

In this manuscript Luthold et al. describe how extracellular matrix proteins and mechanosensation affect melanocyte differentiation. In particular, they show that ECM proteins and surface stiffness lead to effects on the MEK/ERK pathway, thus affecting the MITF transcription factor. The

manuscript is interesting, well written and the data presented in a clear and easy-to-follow manner. The data are nicely quantitated and largely convincing.

1. However, the discussion of the nuclear location of MITF (Figure 4A) is not convincing. The images presented show that upon exposure to Coll, there is a lot of MITF in the nucleus, a lot less so upon CollIV and none upon FN exposure. However, we only see a snapshot of the cells and thus we do not know if we are witnessing effects on MITF protein synthesis, degradation or nuclear localization (the least likely scenario since M-MITF, the isoform present in melanocytes is predominantly nuclear anyway). Was there a cytoplasmic signal detected? Upon FN treatment, there is no MITF protein visible in the cells. Does this mean that the protein is not made, that it is degraded or present at such low levels that the antibody does not detect it? The claim of the authors that this affects nuclear localization of MITF needs more corroboration.

We thank the reviewer for raising this important point regarding the interpretation of MITF localization. We agree that the data as represented in the original figure 4 cannot distinguish whether changes reflect differences in MITF expression, stability, or subcellular distribution.

To better address this, we now included a quantitative analysis of both nuclear and cytoplasmic MITF signals (**revised fig. 4B**). These data show that MITF is detectable in both compartments at all conditions tested. While total MITF levels were not reduced on FN, nuclear MITF was markedly decreased and cytoplasmic MITF was even increased compared to COL I. This indicates that the reduced nuclear signal on FN compared to COL I is not due to an overall loss of MITF protein but rather reflects a shift in its subcellular distribution. These findings support the idea that ECM composition influences MITF localization, consistent with functional changes in its activity and with the observed phenotypic changes.

2. Also, the authors need to show immunocytochemical images for the effects on MITF nuclear localization for the images presented in Figure 5C.

As requested, we now provide representative micrographs illustrating the effects on MITF nuclear localization corresponding to the conditions shown in Fig. 5C. These images have been included in the revised version of the manuscript (**new fig. 6G**), further supporting the quantitative data presented.

3. It seems that the authors quantitated immune-reactivity for both MITF and YAP. What was the control and how was the data normalized?

MITF and YAP immunodetection were performed in separate experiments and were not analyzed in the same cells. For both stainings, secondary antibody controls were included (secondary antibody alone without primary antibody), which showed no detectable signal. For MITF and YAP quantifications (**revised fig. 4B,F**), nuclear (for both) and cytoplasmic

(for MITF) intensity values were normalized within each independent experiment by dividing each individual measurement by the mean nuclear intensity across all conditions. This approach allowed us to deal with total signal variability between experiments while preserving relative differences between ECM conditions. For the percentage of nuclear MITF no normalization was applied. We have added this description to the revised methods section.

4. Similarly, the blots and data shown in Figure 5 are not consistent with the text as described in the results section. The differences observed are minor and the only set that is likely to be significant is the FN-set; the differences between soft, intermediate and stiff of the FN-set do not look significantly different. The description of this in the results section should be toned down accordingly.

To strengthen the conclusions drawn from the original Fig. 5 (**now fig. 6**), we performed additional immunoblot experiments to increase the number of replicates. These extended results now show a statistically significant increase in pERK levels in MCs cultured on FN compared to COL I. However, consistent with the reviewer's observation, no significant differences were detected across the stiffness conditions within FN. We have revised the Results section accordingly to tone down the interpretation and to better reflect the revised data (**revised fig. 6E, lines 339-355**).

Significance:

Upon improvement, this paper will provide an early characterization of the effects of the ECM on melanocyte differentiation. If the link to MITF holds, this will be the first time that mechanosensation has been shown to mediate effects on this transcription factor.

Dear Sandra,

Thank you for submitting your revised manuscript. It has now been seen by all of the original referees.

As you will see, referees find that the study is significantly improved during revision and recommend publication. However, the editorial points below need to be addressed before I can accept the manuscript.

- Please address the remaining concern of referee #1 by toning down the conclusions. Please mark the changes you make in the text.
- Please resubmit the manuscript text as a word file without main and suppl. figures and tables.
- Please reduce the number of keywords to 5.
- Please move the Data Availability section before the Acknowledgements. The Data Availability section is reserved for the primary datasets generated in the study. Each dataset should be listed under a separate bullet point that includes 1) a short description of the measurement type (eg RNA-Seq, ChIP-Seq, mass spectrometry proteomics, imaging, etc...), 2) the name of the repository (or its recommended acronym, see table below and consult fairsharing.org); 3) the DOI or accession number of the dataset; and 4) a resolvable link to the dataset, either in the form of a resolvable link from <http://identifiers.org> or as the full URL to the respective database record. The rest of the statements should be removed from the section.
- Please make the dataset GSE297747 publicly available and include it in the Data Availability section (please see the point above).
- Please rename the Conflicts of Interest section as Disclosure and Competing Interests Statement.
- Please remove the Author Contributions section from the manuscript text.
- We note the following regarding the reference list format: et al needs to be used after 10 author names; DOIs should only be used for preprints and datasets that have not been published yet.
- Please fill out and include an author checklist as listed in our online guidelines (<https://www.embopress.org/page/journal/14693178/authorguide>).
- Please enter complete funding information into the Acknowledgements section of the manuscript, as well as the relevant section of the manuscript submission system.
- Main figures and EV Figures need to be provided as production quality individual Figure files and their legends should be provided in the manuscript text. The suppl. figures should be renamed as EV figures, the nomenclature is Figure EV1, etc. Supplementary nomenclature should not be used.
- During our routine image checks, we noticed that the microscopy panels across the figure set appear pixelated. This is a common result of converting original 16-bit TIFF images to RGB format for publication, and while not a cause for concern, it can sometimes give the impression of image alteration to critical readers. To avoid any misunderstanding and to meet EMBO Press standards, we kindly ask that you: * Resubmit the complete figure set at the captured original data resolution.
- Suppl. Table S1 and S2 provided in the manuscript - these should be removed from the text. We note that Supplementary Table S1 should be converted into Reagents and Tools table (please see the below point) and uploaded separately. We note that the rest of the tables present source data, which should be removed from the manuscript text as well. Source data should be uploaded separately as one zipped file per main figure. Please provide the numerical data in excel format. Please refer to the email sent separately for source data preparation/submission.
- All research articles submitted as revised versions must include a structured methods section that includes a Reagents and Tools Table followed by a Methods and Protocols section. Please see <https://www.embopress.org/page/journal/14693178/authorguide#structuredmethods> for further information.
- Word count, display items should be removed from the manuscript.
- Our production/data editors have asked you to clarify several points in the figure legends - Figure Legends (main + EV):
 - o Please note that the exact p values are not provided in the legends of figures 1B, C, E, F, G; 2C, D; 3B-F; 4B, C, D, F; 6E, F, H, I, K, L, N, O; S3 A-E; S4A, S6B, C.
 - o Please indicate the statistical test used for data analysis in the legend of figure S5 C-E
 - o Please note that information related to n is missing in the legend of figure S5 C.
- Papers published in EMBO Reports include a 'synopsis' and 'bullet points' to further enhance discoverability. Both are displayed on the html version of the paper and are freely accessible to all readers. The synopsis includes a short standfirst summarizing the study in 1 or 2 sentences (max 35 words) that summarize the paper and are provided by the authors and streamlined by the handling editor. I would therefore ask you to include your synopsis blurb and 3-5 bullet points listing the key experimental findings.
- In addition, please provide an image for the synopsis. This image should provide a rapid overview of the question addressed in the study but still needs to be kept fairly modest since the image size cannot exceed 550 (width) x 300-600 (height) pixels.

Thank you again for giving us to consider your manuscript for EMBO Reports, I look forward to your minor revision.

Kind regards,

Deniz

--

Deniz Senyilmaz Tiebe, PhD
Senior Scientific Editor
EMBO Reports

Referee #1:

The authors have addressed my previous concerns. However, their interpretation of the effects of the ECM proteins on MITF nuclear/cytoplasmic localisation is too strong and should be toned down. They see a very mild effect on this - quantitative and not qualitative. This needs to be stressed.

Referee #2:

In the given manuscript, authors investigated how different extracellular matrix proteins and substrate stiffness influence melanocyte behavior, differentiation and mechanosensation. It also highlights the role of MITF as a key regulator affected by these external cues.

Overall, the authors have satisfactorily addressed all the concerns and suggestions raised in the previous review. The revised manuscript reflects the necessary improvements and clarifications. I find the updated version to be scientifically sound, well structured, and suitable for publication in the current form.

Referee #3:

In their revised manuscript, Luthold et al have improved the characterization of the molecular phenotypes observed in mouse melanocytes cultured on different types of matrices. In particular, they have provided additional evidence to consolidate the implication of MITF, document transcriptional changes through RNA sequencing, and clarify the role of FAK. The authors have thus addressed most of the major comments pertaining to the molecular mechanisms underlying the observed cellular phenotypes. They have also expanded the discussion about the in vivo relevance of their findings.

Text and figures are well presented and organized.

The revised version of the manuscript is suitable for publication.

Referee #1:

The authors have addressed my previous concerns. However, their interpretation of the effects of the ECM proteins on MITF nuclear/cytoplasmic localisation is too strong and should be toned down. They see a very mild effect on this - quantitative and not qualitative. This needs to be stressed.

Referee #2:

In the given manuscript, authors investigated how different extracellular matrix proteins and substrate stiffness influence melanocyte behavior, differentiation and mechanosensation. It also highlights the role of MITF as a key regulator affected by these external cues.

Overall, the authors have satisfactorily addressed all the concerns and suggestions raised in the previous review. The revised manuscript reflects the necessary improvements and clarifications. I find the updated version to be scientifically sound, well structured, and suitable for publication in the current form.

Referee #3:

In their revised manuscript, Luthold et al have improved the characterization of the molecular phenotypes observed in mouse melanocytes cultured on different types of matrices. In particular, they have provided additional evidence to consolidate the implication of MITF, document transcriptional changes through RNA sequencing, and clarify the role of FAK. The authors have thus addressed most of the major comments pertaining to the molecular mechanisms underlying the observed cellular phenotypes. They have also expanded the discussion about the in vivo relevance of their findings.

Text and figures are well presented and organized.

The revised version of the manuscript is suitable for publication.

Rev_Com_number: RC-2024-02729

New_manu_number: EMBOR-2025-62056V1-T

Corr_author: Iden

Title: Melanocyte differentiation and mechanosensation are differentially modulated by distinct extracellular matrix proteins

Point by point response to minor revision - EMBOR-2025-62056V1-T

- Please address the remaining concern of referee #1 by toning down the conclusions. Please mark the changes you make in the text.

Referee #1:

The authors have addressed my previous concerns. However, their interpretation of the effects of the ECM proteins on MITF nuclear/cytoplasmic localisation is too strong and should be toned down. They see a very mild effect on this - quantitative and not qualitative. This needs to be stressed.

In response to the remaining concern of Referee #1, we revised the relevant sections (shortened abstract line 50, Results line 250-256 and Discussion line 491; highlighted) to tone down the conclusions as requested.

- Please resubmit the manuscript text as a word file without main and suppl. figures and tables. **COMPLETED**

- Please reduce the number of keywords to 5. **COMPLETED**

- Please move the Data Availability section before the Acknowledgements. **COMPLETED**

The Data Availability section is reserved for the primary datasets generated in the study. Each dataset should be listed under a separate bullet point that includes 1) a short description of the measurement type (eg RNA-Seq, ChIP-Seq, mass spectrometry proteomics, imaging, etc...), 2) the name of the repository (or its recommended acronym, see table below and consult fairsharing.org); 3) the DOI or accession number of the dataset; and 4) a resolvable link to the dataset, either in the form of a resolvable link from <http://identifiers.org> or as the full URL to the respective database record. The rest of the statements should be removed from the section. **COMPLETED**

- Please make the dataset GSE297747 publicly available and include it in the Data Availability section (please see the point above). **COMPLETED**

- Please rename the Conflicts of Interest section as Disclosure and Competing Interests Statement. **COMPLETED**

- Please remove the Author Contributions section from the manuscript text. **COMPLETED**

- We note the following regarding the reference list format: et al needs to be used after 10 author names; DOIs should only be used for preprints and datasets that have not been published yet. **COMPLETED**

- Please fill out and include an author checklist as listed in our online guidelines (<https://www.embopress.org/page/journal/14693178/authorguide>). **COMPLETED**

- Please enter complete funding information into the Acknowledgements section of the manuscript, as well as the relevant section of the manuscript submission system. **COMPLETED**

- Main figures and EV Figures need to be provided as production quality individual Figure files and their legends should be provided in the manuscript text. **COMPLETED**

The suppl. figures should be renamed as EV figures, the nomenclature is Figure EV1, etc. Supplementary nomenclature should not be used. **COMPLETED**

- During our routine image checks, we noticed that the microscopy panels across the figure set appear pixelated. This is a common result of converting original 16-bit TIFF images to RGB format for publication, and while not a cause for concern, it can sometimes give the impression of image alteration to critical readers. To avoid any misunderstanding and to meet EMBO Press standards, we kindly ask that you: * Resubmit the complete figure set at the captured original data resolution.

Thank you for pointing this out, we appreciate your careful check. The pixelation was due to compression in the formal PDF version, which altered the image quality. We have now resubmitted all figures as high-resolution .tiff files at the original data resolution to meet EMBO Press standards, and we trust this resolves the issue.

- Suppl. Table S1 and S2 provided in the manuscript - these should be removed from the text. **COMPLETED**

We note that Supplementary Table S1 should be converted into Reagents and Tools table (please see the below point) and uploaded separately. **COMPLETED**

We note that the rest of the tables present source data, which should be removed from the manuscript text as well. **COMPLETED**

Source data should be uploaded separately as one zipped file per main figure. **COMPLETED**

Please provide the numerical data in excel format. **COMPLETED**

Please refer to the email sent separately for source data preparation/submission. **COMPLETED**

- All research articles submitted as revised versions must include a structured methods section that includes a Reagents and Tools Table followed by a Methods and Protocols section. Please see <https://www.embopress.org/page/journal/14693178/authorguide#structuredmethods> for further information. **COMPLETED**

- Word count, display items should be removed from the manuscript. **COMPLETED**

- Our production/data editors have asked you to clarify several points in the figure legends - Figure Legends (main + EV):

- o Please note that the exact p values are not provided in the legends of figures 1B, C, E, F, G; 2C, D; 3B-F; 4B, C, D, F; 6E, F, H, I, K, L, N, O; S3 A-E; S4A, S6B, C. **COMPLETED**

- o Please indicate the statistical test used for data analysis in the legend of figure S5 C-E. **COMPLETED**

- o Please note that information related to n is missing in the legend of figure S5 C. **COMPLETED**

We have revised the manuscript and added the requested information on exact p-values in the corresponding figure legends. As many conditions are compared, the figure legends have become rather dense. To ensure clarity, we also compiled all statistical details in a dedicated

table within the Source Data (available for each subcaption). If you prefer, we would be happy to streamline the legends by keeping only a reference (e.g. “statistical details available in Source Data”) and defer the full information to the Source Data, in line with EMBO Press standards. We leave this to your judgment.

- Papers published in EMBO Reports include a 'synopsis' and 'bullet points' to further enhance discoverability. Both are displayed on the html version of the paper and are freely accessible to all readers. The synopsis includes a short standfirst summarizing the study in 1 or 2 sentences (max 35 words) that summarize the paper and are provided by the authors and streamlined by the handling editor. I would therefore ask you to include your synopsis blurb and 3-5 bullet points listing the key experimental findings.

Distinct ECM components remodel melanocyte phenotype, mechanosensation and transcriptomic program, at least partly through MITF modulation, offering new insight into how extrinsic cues shape melanocyte differentiation and plasticity.

- MITF has been identified as an ECM- and mechanosensitive transcription factor in melanocytes.
- Collagen I limits MEK/ERK activity, promoting high nuclear MITF and triggering a differentiated, pigmented, proliferative phenotype.
- Fibronectin enhances MEK/ERK signaling, reducing nuclear MITF and melanin production, while inducing a dedifferentiated, motile phenotype.
- On fibronectin, FAK-driven ERK activation suppresses melanogenesis and MEK/ERK inhibition restores both MITF nuclear localization and melanogenesis.
- Melanocyte responses to substrate stiffness are ECM-dependent, highlighting the context-specificity of mechanosensation.

- In addition, please provide an image for the synopsis. This image should provide a rapid overview of the question addressed in the study but still needs to be kept fairly modest since the image size cannot exceed 550 (width) x 300-600 (height) pixels. **COMPLETED**

Prof. Sandra Iden
Saarland University
Cell and Developmental Biology
Kirrberger Str. 100
Homburg 66421
Germany

Dear Sandra,

Thank you for submitting your revised manuscript. I have now looked at everything and all is fine. Therefore, I am very pleased to accept your manuscript for publication in EMBO Reports.

Congratulations on a nice work!

Kind regards,

Deniz

--

Deniz Senyilmaz Tiebe, PhD
Senior Scientific Editor
EMBO Reports
